# Epigenetic profiling of hematopoietic stem cells from male mice identifies KDR and PU.1 as regulators of aging transcriptome and caloric restriction response

Le Zong, Bongsoo Park, Ferda Tekin-Turhan ⓞ, Wakako Kuribayashi, Mayuri Tanaka-Yano ⓞ, Keefer Li & Isabel Beerman ⓞ ✉

Caloric restriction (CR) provides anti-aging benefits but has also been reported to be associated with reduced immune function, and how hematopoietic stem cells (HSCs) potentially contribute to this decline remains unclear. Using life-long and short-term CR in male mice, we found reducing the energy supply decreases total white blood cell production and shifts hematopoiesis towards myeloid and thrombo-erythroid lineages, prioritizing cells essential for survival (red blood cells, platelets, innate immune cells) over adaptive immunity. HSCs under CR enter cell cycle to support myeloid differentiation rather than self-renewal. Lifelong CR inhibits age-associated transcriptome changes in HSCs, though age-associated profiles appear shortly after ad libitum feeding. Epigenetic profiling identified KDR as a key CR response regulator, and *Kdr* knockdown in aged HSCs recapitulated the youthful transcriptome of lifelong CR HSCs. Finally, we show PU.1 acts as an intracellular regulator of CR response, controlling HSC self-renewal and differentiation through increased target gene binding under CR conditions.

Caloric restriction (CR) is defined as a significant reduction in caloric intake without malnutrition (commonly applied as 20% – 40% reduction) and has been reported to extend lifespan and delay age-associated diseases in multiple species[1]. Experimental paradigms using different levels of restriction, timing of the restriction, and composition of the diets show that each of these parameters can contribute to some extent of positive benefits, but overall dietary restriction is associated with mostly beneficial effects on aging phenotypes[2]. C57BL/6J male mice on 40% CR showed increased median and maximum lifespan, whereas this lifespan extension was not seen in female C57BL/6J[3]. 40% CR on male C57BL/6 J mice also improved metabolic efficiency[4] and 30% reduction in these male mice was associated with more robust metabolic flexibility and carbohydrate utilization[5]. While this level of restriction in C57BL/6 males mitigated the loss of splenic CD8+ T-cells in aged mice, with increased

cycling after stimulation (compared to old ad libitum fed)[6], CR has also been associated with suppressed immune response[7–9].

Aging of the hematopoietic system (which is responsible for immune response) is associated with decreases in lymphoid potential, bias toward myeloid cell generation, increased inflammation, and increased susceptibility to infection and disease. As tissue-specific stem cells are responsible for the maintenance and repair of their system throughout life, age-associated changes in the hematopoietic stem cell (HSC) compartment are thought to contribute to these hematopoietic alterations. While CR studies have shown positive benefits on various other adult tissue-specific stem cell compartments such as intestinal stem cells, skeletal muscle stem cells, and neural stem and progenitor cells during aging[10–13], manuscripts examining the effect of dietary restriction on HSCs have conflicting findings[9,14,15]. Variations in these studies included the duration of caloric restriction

Translational Gerontology Branch, National Institute on Aging, Baltimore, MD, USA. ✉e-mail: isabel.beerman@nih.gov

(9-21 months), the level of restriction (30%-40%), and potentially the sex of the animals.

As positive life- and health-span benefits have been consistently reported at 40% CR in C57BL/6 J males, we sought to investigate this level of lifelong dietary restriction on hematopoiesis, with a focus on the aged HSC compartment's response to this sustained stress to understand the molecular signatures that may arise contributing to global effects on the blood system.

## Results

### Reduced white blood cell production in lifelong CR male mice

To investigate the impact of CR on the hematopoietic system during aging, C57BL/6 J male mice were subjected to lifelong 40% CR (60% of ad libitum diet) starting at the age of 3−4 months (Fig. 1a). Additionally, short-term CR (4 weeks) was performed to examine the effects of CR without the influence of aging (Supplementary Fig. 1a). Complete blood count analysis of peripheral blood (PB) isolated from old mice (24−26 months) showed lifelong CR (OCR, Old Caloric Restricted) resulted in a significant reduction in white blood cell (WBC) production compared to lifelong ad libitum (ad lib) access to chow (OAL, Old Ad Libitum) (Fig. 1b). We also evaluated the response to 4 weeks ad lib access to food after lifelong CR (OCRRF, Old Caloric Restricted Re-Fed) and found WBC production restored to levels similar to the OAL mice (Fig. 1b). Lifelong CR affected the production of both myeloid (neutrophil) and lymphoid (lymphocyte) PB cells, with a larger reduction in lymphocyte numbers (Fig. 1c, d). Consistently, the weight of the spleen, an organ that produces lymphocytes, was significantly reduced in OCR mice and recovered after refeeding (Fig. 1e). These changes in PB parameters were mirrored in the short-term CR (YCR, Young Caloric Restricted) results (Supplementary Fig. 2a−d). Age associated changes were also observed, with reduction in red blood cells and increase in platelets in ad lib fed old mice compared to ad lib fed young mice (Fig. 1f, g). However, the number of red blood cells and platelets in PB were maintained under CR conditions (Fig. 1h, i).

### Biased differentiation towards myeloid cells in lifelong CR mice

To further examine changes in the frequency of specific blood cell types, we analyzed blood cell frequencies using flow cytometry (Supplementary Fig. 1b). The results showed an increase in Mac-1[+] myeloid cells (Fig. 1j and Supplementary Fig. 2e), particularly granulocytes (Fig. 1k and Supplementary Fig. 2f), and a decrease in B cells in both lifelong and short-term CR mice (Fig. 1l and Supplementary Fig. 2g). The frequency of T cells was largely maintained, although the absolute number of T cells was reduced under both short-term and lifelong CR (Supplementary Fig. 2h, i). In contrast, CR decreased both the absolute number and frequency of B cells (Fig. 1l, Supplementary Fig. 2g and j), suggesting that B cells are more selectively suppressed than T cells under CR conditions. Overall, the mice maintained a myeloid-biased PB composition under CR conditions (Supplementary Fig. 2k). Analysis of whole bone marrow (WBM) cells revealed an increased frequency of megakaryocyte-erythroid progenitors (MEP) in CR mice (Fig. 1m, and Supplementary Fig. 1c and 3a), consistent with the maintained red blood cell and platelet production in PB (Fig. 1h, i). The frequencies of common myeloid progenitors (CMP) and granulocyte-monocyte progenitors (GMP) remained unchanged under CR conditions (Supplementary Fig. 3b, c). However, the frequency of common lymphoid progenitors (CLP) was reduced (Fig. 1n and Supplementary Fig. 3d), along with decreased frequencies of Lin⁻Sca-1⁺c-Kit⁺ (LSK) cells and MPPFlk2⁺ cells (lymphoid-biased multipotent progenitors) in CR mice (Fig. 1o, p and Supplementary Fig. 3e, f), consistent with the reduced lymphoid output seen in the PB (Supplementary Fig. 2k). Interestingly, lifelong CR ameliorated age-dependent HSC expansion (Fig. 1q and Supplementary Fig. 3g), a well-defined HSC aging phenotype[16]. CR also altered HSC composition, with a decreased frequency of CD150[high]

(myeloid-biased) HSCs and maintenance of CD150[low] (balanced lineage output) HSCs (Supplementary Fig. 3h). Upon refeeding, the frequency of CD150[high] HSCs returned to ad lib levels, whereas CD150[low] HSC frequency decreased (Supplementary Fig. 3h). These results suggest that under CR, myeloid-biased CD150[high] HSCs are preferentially consumed to sustain myeloid differentiation, while CD150[low] HSCs contribute to the burst of lymphoid differentiation upon refeeding. Importantly, overall bone marrow cellularity remained largely unchanged after CR (Supplementary Fig. 3i), in contrast to the peripheral blood, where total white blood cell counts were significantly reduced under CR (Supplementary Fig. 2a). This distinction highlights a key difference between the bone marrow and peripheral blood in their responses to CR. Together, our data indicate that myeloid differentiation is preferentially maintained, while lymphoid differentiation is repressed under CR conditions, despite an overall reduction in white blood cell output (Supplementary Fig. 3j). These results suggest that, under CR stress, male mice modify their differentiation programs to prioritize the production of cell types essential for survival.

### CR drives HSCs into cell cycle to maintain myeloid differentiation rather than self-renewal

We then performed RNA-seq on HSCs (LSK CD34⁻Flk2⁻CD150⁺) isolated from both lifelong and short-term CR mice to investigate how CR regulates HSC function. Principal component analysis (PCA) showed that samples from the same treatment group clustered together, and samples from the refeeding group were closer to the ad lib group (Fig. 2a and Supplementary Fig. 4a). Differentially expressed gene (DEG) analysis identified similar number of up- and down-regulated genes (up 897 vs down 743) in OCR HSCs (Fig. 2b, and Supplementary Data 1). However, although the overall number of DEGs in YCR HSCs was lower compared to OCR HSCs, we observed a significant bias toward down-regulation of transcripts in YCR versus YAL (31 up vs 105 down of 136 DEGs; two-sided exact binomial test, $p = 1.33 \times 10^{-10}$) (Supplementary Fig. 4b, Supplementary Data 1). The transcriptome largely reverted to the levels of similarly aged ad lib HSCs after refeeding for both YCR and OCR groups (Fig. 2b and Supplementary Fig. 4b). Surprisingly, pathway analysis revealed that upregulated genes in OCR HSCs were significantly enriched in cell cycle related pathways (Fig. 2c and Supplementary Fig. 4c). Gene Set Enrichment Analysis (GSEA) confirmed that OCR HSCs were enriched for the expression of cell cycle and checkpoint related genes (Fig. 2d). Analysis of short-term CR data showed that YCR HSCs were also enriched for the expression of cell cycle and checkpoint related genes, although to a lesser extent compared to OCR HSCs (Supplementary Fig. 4d). Direct examination of cell cycle (including *Mki67*, *Cdk1*, and *Ccna2*) and checkpoint (including *Chek1*, *Chek2*, and *Brca1*) related transcripts confirmed their upregulation in OCR HSCs (Fig. 2e, f and Supplementary Fig. 4e, f). These results were unexpected as CR led to an overall reduction in white blood cell production and reduced age-dependent HSC expansion (Fig. 1b, q). To confirm HSCs were indeed cycling more under CR conditions, we restricted young Ki67-RFP reporter mice with a CR diet for three months then examined the ratio of Ki67⁺ cycling HSCs (Supplementary Fig. 4g). The result showed that CR led to an ~2-fold increase in the ratio of actively cycling HSCs compared to ad lib feeding (Supplementary Fig. 4h). We then investigated whether the increased cycling HSCs were involved in self-renewal or differentiation. CellRadar (software to visualize cell-type gene enrichment based on normal mouse or human hematopoiesis expression) analysis of up- and down-regulated genes in OCR HSCs showed downregulated genes were mainly enriched for long-term HSC (LT-HSC) related genes, while the upregulated genes were enriched for short-term HSC (ST-HSC), granulocyte progenitor (pre-GM and GMP), and erythrocyte progenitors (CFUE, ProE) (Fig. 2g). In short-term CR mice, the downregulated genes in YCR HSC were also mainly LT-HSC

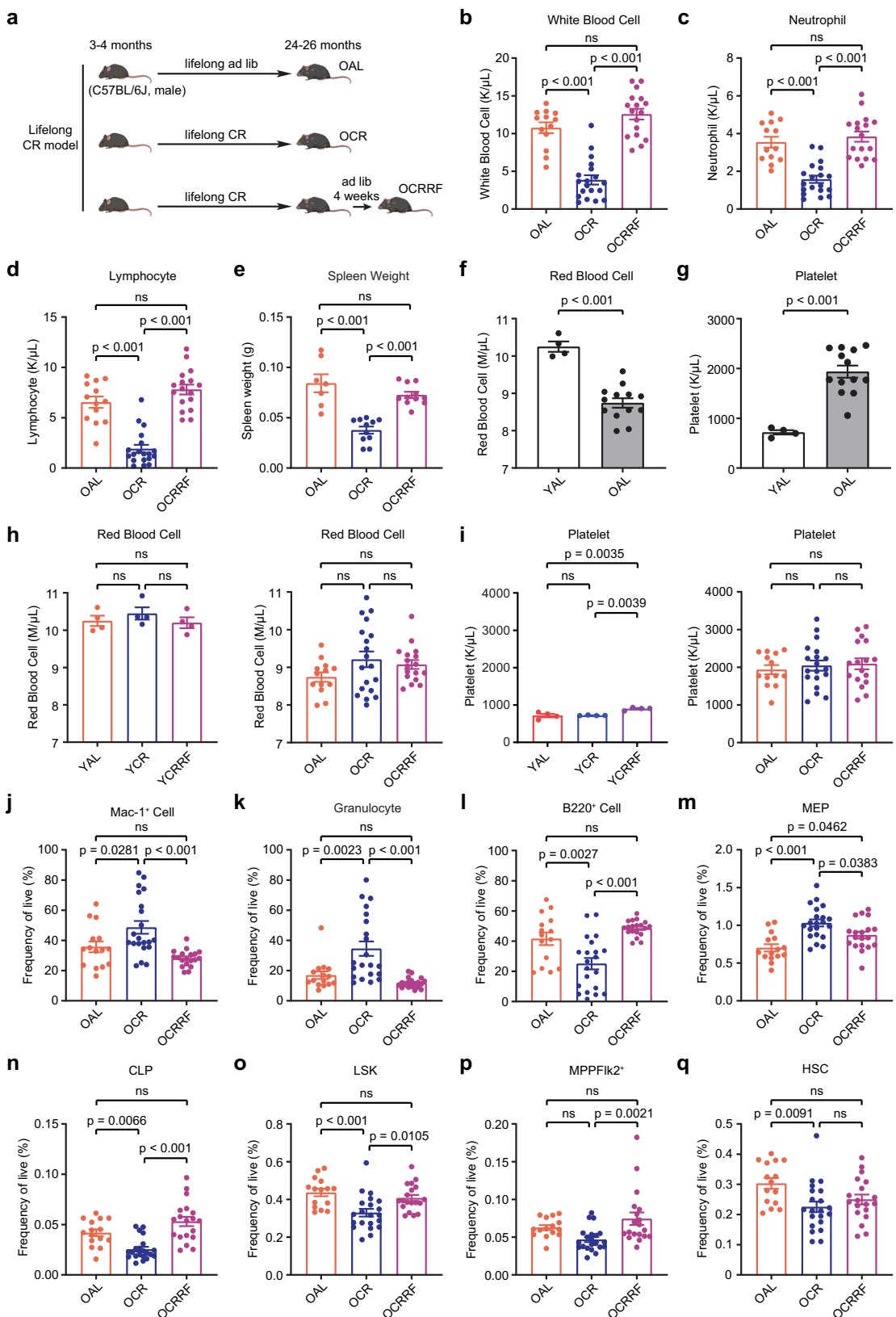

related genes, and the upregulated genes were specifically enriched in ST-HSC and pre-GM (Supplementary Fig. 4i). These results suggest under CR stress, HSCs enter cell cycle and preferentially express transcripts directing differentiation over self-renewal. Together, our data suggest that CR drives HSCs into the cell cycle to maintain myeloid differentiation rather than self-renewal.

## Lifelong CR preserves a youthful transcriptome in aged HSCs

We next investigated whether CR explicitly affects established age-associated changes in the HSC transcriptome. Towards this, we used the previously defined HSC aging signature (AS) gene list, a consensus group of differentially expressed transcripts identified by at least four independent studies[17]. Our results showed that lifelong CR significantly

**Fig. 1 | Lifelong CR leads to reduced white blood cell production and myeloid-biased differentiation. a** Lifelong CR paradigm. OAL (Old Ad Libitum), lifelong ad libitum feeding; OCR (Old Caloric Restricted), lifelong CR; OCRRF (Old Caloric Restricted Re-Fed), 4 weeks ad libitum access to food after lifelong CR. Created in BioRender. Ma, F. (https://BioRender.com/fdntz9h). **b**–**d** Complete blood count of white blood cells (**b**), neutrophils (**c**), and lymphocytes (**d**) in OAL (87 = 13), OCR (n = 19), and OCRRF (n = 17) mice. Data are represented as mean ± SEM, one-way ANOVA. Source data are provided as a Source Data file. **e** Spleen weight of OAL (n = 7), OCR (n = 11) and OCRRF (n = 11) mice. Data are represented as mean ± SEM, one-way ANOVA. Source data are provided as a Source Data file. **f, g** Complete blood count of red blood cells (**f**) and platelets (**g**) in YAL and OAL mice. Data are represented as mean ± SEM, YAL (n = 4), OAL (n = 13), two-tailed t test. Source data

are provided as a Source Data file. **h, i** Complete blood count of red blood cells (**h**) and platelets (**i**) in young and old CR mice. Data are represented as mean ± SEM, young (n = 4), OAL (n = 13), OCR (n = 19), OCRRF (n = 17), one-way ANOVA. Source data are provided as a Source Data file. **j, l** Frequency of Mac-1⁺ cells (**j**), granulocytes (**k**), and B220⁺ cells (**l**) in the peripheral blood (PB) of OAL (n = 15), OCR (n = 21), and OCRRF (n = 19) mice. Data are represented as mean ± SEM, one-way ANOVA. Source data are provided as a Source Data file. **m**–**q** Frequency of megakaryocyte-erythroid progenitors (MEP) (**m**), common lymphoid progenitors (CLP) (**n**), Lin⁻Sca-1⁺c-Kit⁺ cells (LSK) (**o**), lymphoid-biased multipotent progenitors (MPPFlk2⁺) (**p**), and hematopoietic stem cells (HSCs) (**q**) in the bone marrow of OAL (n = 15), OCR (n = 21), and OCRRF (n = 19) mice. Data are represented as mean ± SEM, one-way ANOVA. Source data are provided as a Source Data file.

---

reduced the expression of AS genes upregulated in aged HSCs and significantly increased the expression of AS genes downregulated in aged HSCs (Fig. 3a and Supplementary Data 2). GSEA analysis confirmed that OCR HSCs exhibited lower expression of upregulated AS genes and were enriched for higher expression of downregulated AS genes (Fig. 3b). PCA revealed that lifelong CR shifted the transcriptome toward the young HSC state along PC1 (Supplementary Fig. 5a). Unsupervised clustering analysis showed that YAL and YCR grouped together, while OAL and OCR formed a separate cluster (Supplementary Fig. 5b). Nonetheless, the transcriptomic profile of OCR HSCs clearly shifted toward - but did not fully converge with - the YAL state, indicating a partial transcriptional reversion under CR. Examining the AS genes which were differentially expressed in the opposite direction in OCR compared to OAL HSCs showed that nearly half of the AS genes exhibited a significant mitigation in expression changes under CR conditions (Fig. 3c). This included *Clu* and *Osmr*, which are upregulated HSC aging signature genes (Fig. 3d). *Clu* (clusterin) encodes a secreted glycoprotein that drives myeloid bias in aged HSCs by regulating mitochondrial function[18], while *Osmr* (oncostatin M receptor) encodes a cytokine receptor involved in the maintenance of erythroid and megakaryocyte progenitors[19]. In contrast, *Ect2* and *Mcm5* are downregulated HSC aging signature genes (Fig. 3e). *Ect2* (epithelial cell transforming 2) encodes a guanine nucleotide exchange factor required for cytokinesis and cell cycle progression, whereas *Mcm5* (minichromosome maintenance complex component 5) encodes a subunit of the MCM helicase complex essential for DNA replication initiation and serves as a marker of proliferative potential. Additionally, elevated expression of transposable elements (TEs) has been reported in aged HSCs[20]. In OCR HSCs compared to OAL, we observed mostly downregulation of TE expression (Fig. 3f), again suggesting that lifelong CR helps preserve a more youthful transcriptome in HSCs.

## Epigenetic repression of *Kdr* and *Bmpr1a* expression contributes to the improved transcriptome in OCR HSCs

Given the global changes to transcription, we wanted to screen for altered epigenetic regulation underlying the mitigated HSC aging signature under CR conditions. Most literature focuses on histone acetylation in CR conditions as sirtuin expression is typically altered. However, we did not observe changes in the sirtuin family expression, but we found significant changes in other regulators of histone modifications, particularly those involved with histone methylation (Supplementary Data 3). Thus, we performed histone ChIP-seq on OAL and OCR HSCs on two histone methylation marks H3K4me3 and H3K27me3 which are key regulators of transcription. Though stem cell numbers are limiting in ChIP assays, QC analysis of our samples showed high correlation between replicates for both H3K4me3 (permissive mark) and H3K27me3 (repressive mark) antibodies within treatment groups (Supplementary Fig. 6a). Global comparisons of all histone peaks in OAL and OCR HSCs revealed a general trend of reduced histone modification levels for both H3K4me3 and H3K27me3, with a more pronounced loss for H3K4me3 in the OCR

(Fig. 4a), there were also decreased levels of both H3K4me3 and H3K27me3 at gene promoters (Supplementary Fig. 6b) in OCR HSCs compared to the OAL. To identify the candidate loci that contribute to altered transcription, we analyzed the most differentially modified promoter regions, selecting loci with the greatest relative increase or decrease in ratio of histone peaks in OAL and OCR HSCs (Fig. 4a).

Consistent with the concept that H3K4me3 positively corelates with gene expression and H3K27me3 negatively correlates with gene expression, the genes that were in the top 10% of relative H3K4me3 levels between OCR and OAL (H3K4me3-high) and the loci with the lowest 10% of relative H3K27me3 levels (H3K27me-low) had higher expression levels in OCR HSCs (Supplementary Fig. 6c). Conversely, the H3K4me3-low and H3K27me3-high related genes had lower expression levels in OCR HSCs (Supplementary Fig. 6c). Given that the expression of AS genes trends toward a younger state in OCR HSCs (Fig. 3a, b), we examined whether changes in histone modifications in OCR HSCs are associated with the mitigation of AS gene expression changes in the OCR HSCs. For the 180 upregulated AS genes in aged HSCs[17], 61 genes were associated with relatively lower H3K4me3 in OCR, and 17 genes overlapped with H3K27me3-high genes in OCR HSCs (Supplementary Fig. 6d). For 39 downregulated AS gene[17], 7 genes overlapped with H3K4me3-high genes, and 1 gene overlapped with H3K27me3-low genes in OCR HSCs (Supplementary Fig. 6d). We focused on the 17 AS genes that showed increased levels of H3K27me3 in the OCR HSCs, as these H3K27me3 regions went against the global trend of overall decreased H3K27me3 level suggesting concerted mechanisms to increase the repression at these locations. Further, aged HSCs have global decreases in H3K27me3 compared to young HSCs (manuscript under review), and thus this increase of H3K27me3 might be reflective of a younger epigenetic status. These 17 AS genes also had lower levels of H3K4me3, and almost all showed reduced gene expression (Fig. 4b). We focused on *Kdr* and *Bmpr1a* which showed the highest fold increase in H3K27me3 and also had robust change in gene expression. Increased levels of H3K27me3 and decreased levels of H3K4me3 were at the promoters of *Kdr* and *Bmpr1a*, associated with reduced gene expression in the OCR HSCs compared to the OAL (Fig. 4c, d), suggesting this altered epigenetic regulation contributed to the changes in expression. As both *Kdr* and *Bmpr1a* encode cell-surface receptors involved in multiple signaling pathways, the regulatory genes have potential to globally impact gene transcription, and we felt these would be strong candidates for contributing to altered transcriptional profiles of the OCR HSCs. Additionally, *Bmp4*, the gene coding a ligand for *Bmpr1a*, also showed similar histone modification changes and decreased gene expression in OCR HSCs (Supplementary Fig. 6e).

To explore the role of KDR and BMPR1A in aged HSCs, we explored if knockdown of *Kdr* and *Bmpr1a* in aged HSCs could recapitulate the improved transcriptome signature observed in OCR HSCs. To investigate the effects of combinational and individual knockdowns, we infected purified, aged HSCs with both GFP-labeled *Kdr* shRNA virus (shRNA5) and mCherry-labeled *Bmpr1a* shRNA virus

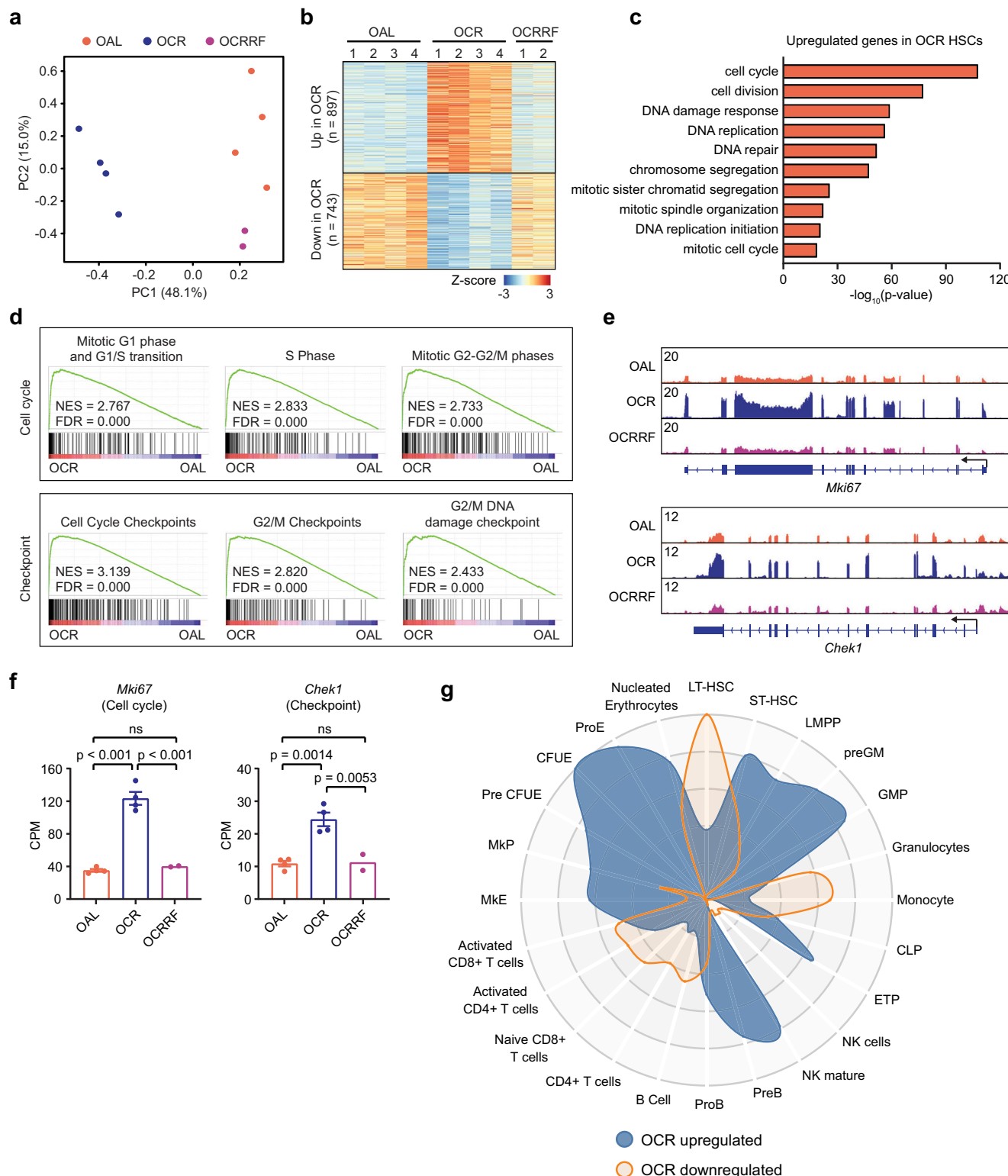

**Fig. 2 | HSCs are driven into the cell cycle to maintain myeloid differentiation rather than self-renewal. a** PCA plot of RNA-seq datasets from OAL ($n = 4$), OCR ($n = 4$), and OCRRF ($n = 2$) HSC samples. **b** Heatmap of differentially expressed genes (DEGs) in OAL versus OCR comparison (FC > 1.2, FDR < 0.05). RNA-seq data of HSCs purified from OAL ($n = 4$), OCR ($n = 4$), and OCRRF ($n = 2$) mice were used. **c** Pathway analysis of upregulated DEGs in OCR HSCs. Gene Ontology enrichment was performed using DAVID. Enrichment p-values correspond to the DAVID EASE Score (modified one-sided Fisher's exact test) without correction for multiple testing. **d** GSEA analysis of cell cycle and checkpoint related pathways using RNA-seq data from OAL and OCR HSCs. **e** Integrative Genomics Viewer (IGV)[52] view of *Mki67* and *Chek1* expression using RNA-seq data from OAL, OCR, and OCRRF HSCs. **f** Expression levels of *Mki67* and *Chek1* in OAL ($n = 4$), OCR ($n = 4$), and OCRRF ($n = 2$) HSCs. Data are represented as mean ± SEM, one-way ANOVA. **g** CellRadar plot derived from DEGs of OAL versus OCR comparison (FC > 1.2, FDR < 0.05). RNA-seq data of HSCs purified from OAL ($n = 4$), OCR ($n = 4$).

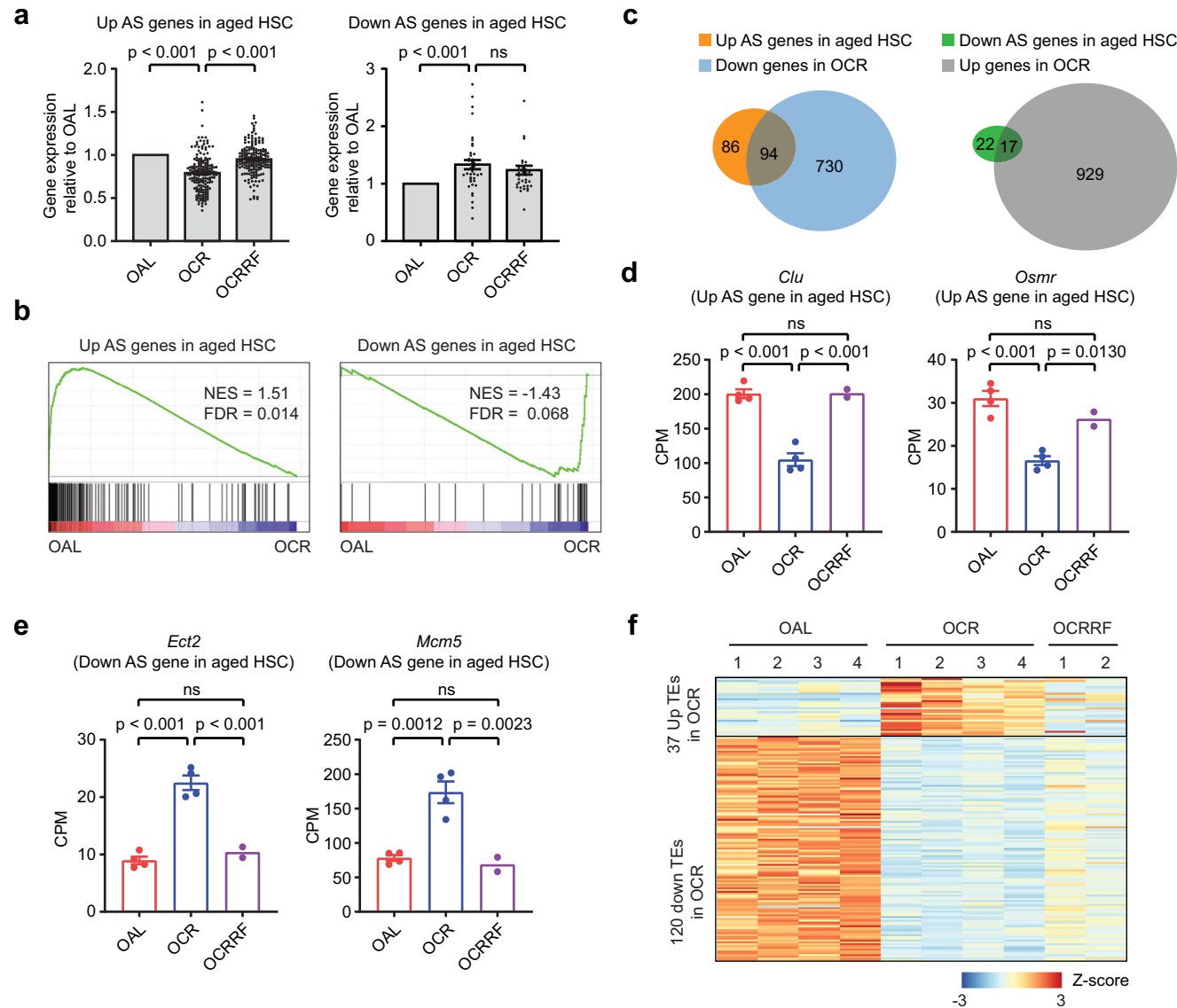

**Fig. 3 | Lifelong CR preserves a youthful transcriptome in OCR HSCs. a** Relative expression levels of aging signature (AS) genes in OAL, OCR, and OCRRF HSCs. List of upregulated (left, $n = 180$) and downregulated (right, $n = 39$) AS genes were analyzed[18]. Data are represented as mean ± SEM, one-way ANOVA. **b** GSEA plots generated with upregulated or downregulated AS genes. RNA-seq data of HSCs purified from OAL ($n = 4$) and OCR ($n = 4$). **c** Venn diagram showing the overlap between AS genes and DEGs from OAL versus OCR comparison. **d**, **e** Expression levels of upregulated (*Clu* and *Osmr*) (**d**) and downregulated (*Ect2* and *Mcm5*) (**e**) AS genes in OAL ($n = 4$), OCR ($n = 4$), and OCRRF ($n = 2$) HSCs. Data are represented as mean ± SEM, one-way ANOVA. **f** Heatmap of differentially expressed transposable elements (TEs) between OAL and OCR HSCs (FC > 1.5, FDR < 0.05).

(shRNA1,2), or both GFP- and mCherry-labeled scramble shRNAs. These cells were then transplanted into lethally irradiated recipient mice with a radioprotective dose of myeloid progenitor cells (Fig. 4e and Supplementary Fig. 6f, g). This transplant paradigm allowed for continuous knockdown of these transcripts in HSCs, not feasible in vitro. At the end of the transplantation period, *Kdr* and *Bmpr1a* double knockdown HSCs, as well as *Kdr* single knockdown HSCs, were sorted for RNA-seq analysis (Fig. 4e). Scramble shRNA infected HSCs (GFP+mCherry+, or GFP+) were purified from transplanted mice as controls. GSEA analysis of RNA-seq data revealed that both the *Kdr* and *Bmpr1a* double knockdown, as well as the *Kdr* single knockdown resulted in decreased expression of upregulated AS genes (Supplementary Fig. 6h, left). Conversely, these knockdowns were enriched for increased expression of downregulated AS genes (Supplementary Fig. 6h, right). The effect was more pronounced with the double knockdown, as indicated by a higher enrichment score

(Supplementary Fig. 6h). These findings suggest that knockdown of *Kdr* and *Bmpr1a* can shift the aging transcriptome of HSCs towards a more youthful state and imply that epigenetic repression of *Kdr* and *Bmpr1a* expression contributes to the improved transcriptome in OCR HSCs.

## Cell surface receptor *Kdr* is a regulator of the CR response in OCR HSCs

When performing GSEA analysis with the DEGs identified in OCR HSCs, we observed that while both the *Kdr* and *Bmpr1a* double knockdown and the *Kdr* single knockdown can mimic the CR-associated gene downregulation in OCR HSCs, only the *Kdr* single knockdown recapitulated the CR-associated gene upregulation (Fig. 4f). A direct comparison of DEG expression levels showed that the DEGs downregulated in OCR HSCs were significantly downregulated in both the double and single knockdown HSCs (Fig. 4g, upper panel). However, DEGs

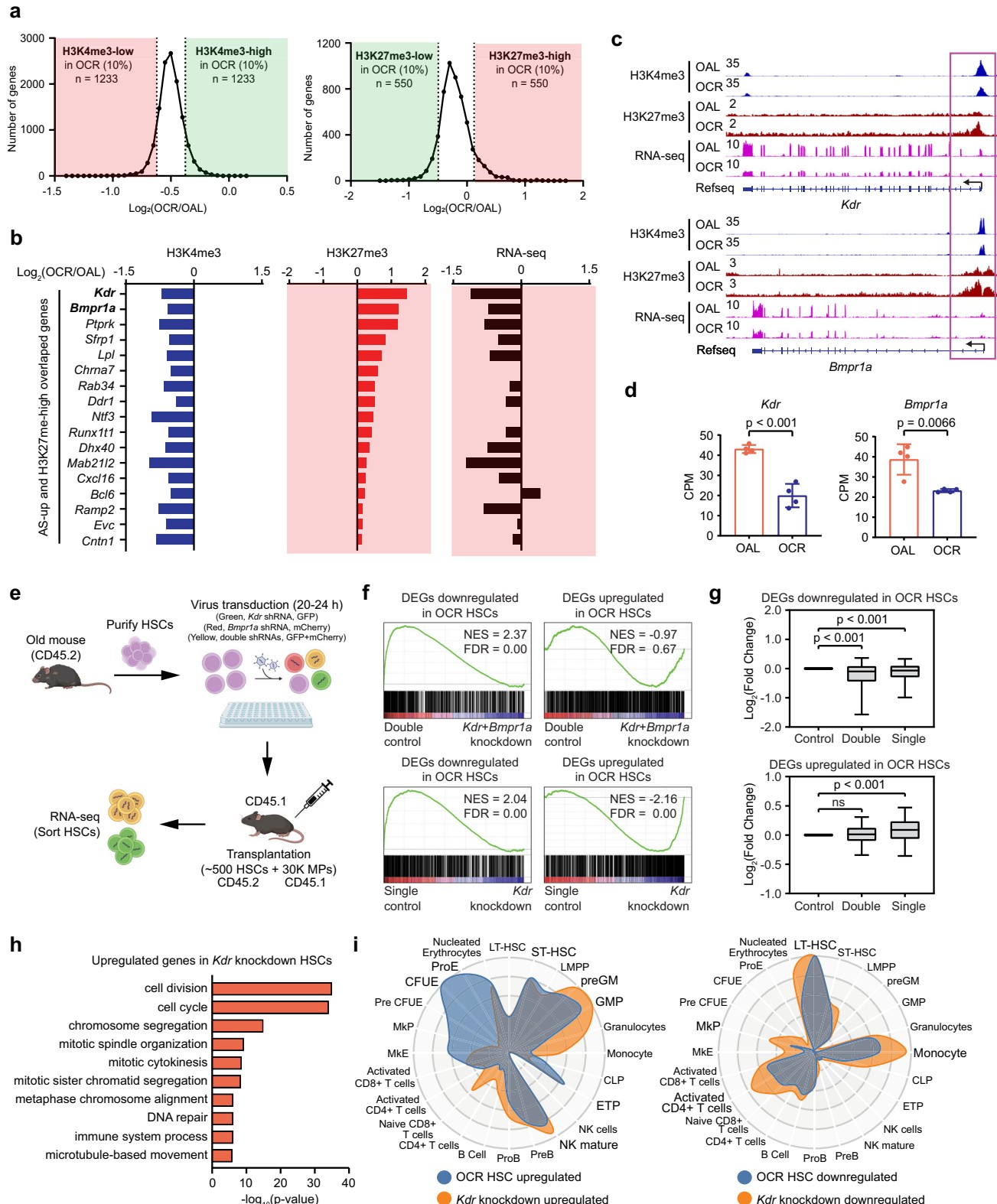

upregulated in OCR HSCs were only significantly upregulated in the *Kdr* single knockdown HSCs (Fig. 4g, lower panel).

Under CR stress, HSCs are driven into cell cycle and maintain circulating myeloid cells (Fig. 2). To investigate whether *Kdr* plays a role in regulating the response of HSCs to CR, we performed DEG analysis using RNA-seq data from control (GFP-Scramble) and *Kdr* knockdown (*Kdr*-shRNA5) HSCs (Supplementary Fig. 6i). Pathway analysis indicated that DEGs in the *Kdr* knockdown HSCs were involved

in cell cycle related pathways (Fig. 4h and Supplementary Fig. 6j). CellRadar analysis showed a clear overlap between expression plots generated with DEGs identified in OCR HSCs compared to OAL HSCs and those from *Kdr* knockdown old HSCs compared to scramble control old HSCs (Fig. 4i). In *Kdr* knockdown HSCs, upregulated DEGs were enriched in genes related to ST-HSC and myeloid/erythroid cells (preGM, GMP, nucleated erythrocytes), while downregulated DEGs were enriched in LT-HSC and monocyte related genes (Fig. 4i). These

**Fig. 4 | KDR is a regulator of aging transcriptome and CR response in HSCs.**
**a** Comparison of H3K4me3 and H3K27me3 levels at gene promoters in OAL and OCR HSCs. For H3K4me3 ChIP-seq, signals within ±1 kb of the transcription start site (TSS) were analyzed. For H3K27me3 ChIP-seq, signals from 0 to 2 kb downstream of the TSS were used for analysis. **b** Relative H3K4me3 and H3K27me3 levels at the promoters of 17 overlapping genes between upregulated AS genes and those categorized as H3K27me3-high in OCR HSCs, compared to their levels in OAL HSCs. The corresponding expression of these 17 genes is shown on the right. **c** IGV tracks displaying H3K4me3, H3K27me3, and gene expression levels at the *Kdr* and *Bmpr1a* loci. **d** Expression levels of replicates were shown on the right. Data are represented as mean ± SEM, *n* = 4, two-tailed t test. **e** Experiment workflow of shRNA mediated knockdown targeting *Kdr* and/or *Bmpr1a* in aged HSCs followed by transplantation. Created in BioRender. Ma, F. (https://BioRender.com/fdntz9h). **f** GSEA plots generated using DEGs from the OAL versus OCR comparison (FC > 1.2, FDR < 0.05).

RNA-seq data from *Kdr* and *Bmpr1a* double knockdown (upper two panels) and *Kdr* single knockdown (lower two panels) HSCs were used. Double knockdown control, *n* = 2; double knockdown, *n* = 4; single knockdown control, *n* = 2; single knockdown, *n* = 4. **g** Box plots showing gene expression changes of DEGs (downregulated, *n* = 743; upregulated, *n* = 897) from OAL versus OCR comparison. RNA-seq data from *Kdr* and *Bmpr1a* double knockdown and *Kdr* single knockdown HSCs were used. Two-tailed t test. Box plots show the median (center line), the 25th and 75th percentiles (box), and the 5th and 95th percentiles (whiskers). **h** Pathway analysis of upregulated DEGs in *Kdr* knockdown HSCs. Gene Ontology enrichment was performed using DAVID. Enrichment p-values correspond to the DAVID EASE Score (modified one-sided Fisher's exact test) without correction for multiple testing. **i** CellRadar plots derived from upregulated DEGs in OCR HSCs and *Kdr* knockdown HSCs (left), and downregulated DEGs in OCR HSCs and *Kdr* knockdown HSCs (right).

findings suggest that *Kdr* knockdown induces a transcriptome that favors cell cycle progression and myeloid differentiation while inhibiting HSC self-renewal, mimicking the effects of CR. This indicates that *Kdr* not only contributes to the improved aging transcriptome in HSC under CR conditions but also plays a crucial role in regulating the CR response in HSCs.

## PU.1 regulates the response of HSCs to CR by increased binding to its target genes

To identify potential intracellular regulators of the CR response we performed ATAC-seq on OAL, OCR, and OCRRF HSCs. PCA analysis revealed that samples from the same treatment group clustered together (Supplementary Fig. 7a). Differentially accessible region (DAR) analysis identified a similar number of DARs with increased (663) or decreased (673) accessibility in OCR HSCs (Fig. 5a and Supplementary Fig. 7b). Notably, the changes in chromatin accessibility were largely reversible, with accessibility in OCRRF HSCs returning towards the levels observed in OAL HSCs (Fig. 5a). Transcription factor motif analysis of these DARs highlighted a significant enrichment of PU.1 and SPIB binding motifs in regions with increased accessibility in OCR HSCs (Fig. 5b). Given the similarity in PU.1 and SPIB binding motifs and the higher expression levels of *Spi1* (the gene encoding PU.1) compared to *SpiB* in HSCs (Supplementary Fig. 7c), our results suggested PU.1 may be a key regulator of the CR response in HSCs.

To test this, we conducted PU.1 ChIP-seq on OAL, OCR, and OCRRF HSCs. Genome browser viewing of our data confirmed the reported self-regulation of PU.1[21], with binding of PU.1 upstream of its promoter (Supplementary Fig. 7d). Consistent with the motif enrichment analysis, we observed increased PU.1 binding in OCR HSCs at DARs with a PU.1 binding motif, which returned to OAL levels upon refeeding (Fig. 5c). Genome-wide comparison of PU.1 binding demonstrated an overall trend of increased binding in OCR HSCs compared to OAL HSCs, which normalized after refeeding (Fig. 5d). Consistent with this, we identified more peaks with increased PU.1 binding (1942) rather than decreased binding (236) in OCR HSCs (FC > 1.5) (Fig. 5e). Comparisons of genes associated with increased PU.1 binding and DEGs in OCR HSCs showed a balanced number of upregulated and downregulated DEGs with increased PU.1 binding (Fig. 5f), consistent with the reported activity of PU.1 functioning as both an activator and repressor of gene transcription[22]. Pathway analysis of these overlapping DEGs revealed an enrichment of cell cycle related pathways among the upregulated DEGs in OCR HSCs (Fig. 5g and Supplementary Fig. 7e), indicating that increased PU.1 binding is involved in regulating the cell cycle response to CR in HSCs. CellRadar analysis further showed that the upregulated overlapping DEGs in OCR HSCs were enriched for erythroid related genes (nucleated erythrocyte), while the downregulated overlapping DEGs were enriched for LT-HSC related genes (Fig. 5h). This suggests that increased PU.1 binding in OCR HSCs is also involved in regulating differentiation

choices and inhibiting HSC self-renewal. To investigate potential binding partners of PU.1, we performed transcription factor motif enrichment analysis using HOMER, examining sequences within ±200 bp of the centers of PU.1 peaks associated with genes either upregulated or downregulated after CR. Both de novo and known motif analyses were conducted. As expected, the PU.1 motif was the most significantly enriched motif in both up- and downregulated gene-associated peaks (Supplementary Fig. 8a), confirming the specificity and robustness of our dataset. Among known motifs, we observed similar enrichment of ETS family transcription factors - including ELF4, ETS1, FLI1, GABPA, and ERG - across both up- and downregulated gene-associated peaks (Supplementary Fig. 8a, left). These factors share the ETS DNA-binding domain and are known to act as either activators or repressors in a context-dependent manner, consistent with PU.1's dual regulatory roles. Interestingly, de novo motif analysis revealed modest enrichment of distinct motifs between the two gene sets: NFIA and HMGA1 motifs were more enriched in PU.1 peaks linked to upregulated genes, whereas ZFP335 and MYBL2 motifs were more enriched in those associated with downregulated genes, although their enrichment p-values were substantially higher than that of PU.1 (Supplementary Fig. 8a, right). These results suggest that PU.1 may engage different co-regulatory partners in distinct transcriptional contexts. Alternatively, PU.1's dual activity may involve sequence-independent mechanisms, such as recruitment of different chromatin modifiers to regulate transcription. To further explore the potential link between KDR and PU.1 pathways, we analyzed whether genes differentially expressed following *Kdr* knockdown were regulated by PU.1. ~50% of these DEGs had PU.1 binding within 50 kb of the transcription start site (TSS) (Supplementary Fig. 8b), indicating substantial potential regulatory overlap. Moreover, chromatin profiling revealed increased PU.1 binding after CR (Supplementary Fig. 8c), suggesting that PU.1 contributes to the transcriptional regulation of KDR-responsive genes under CR conditions. We also found that both CR and *Kdr* knockdown led to a modest increase in *Spi1* expression (Supplementary Fig. 8d), and ChIP-seq data showed PU.1 occupancy near the *Kdr* promoter (Supplementary Fig. 8e), supporting a possible reciprocal regulatory interaction between KDR signaling and PU.1 activity. In summary, our data support a role for PU.1 as an intracellular regulator of the CR response in HSCs and suggest that it may cooperate with KDR signaling to orchestrate the corresponding transcriptional program under CR conditions.

## Discussion

40% CR has been widely used as an effective method to mitigate aging-related functional decline[1]. However, despite its benefits, CR can also cause stress to the organism. Understanding how the hematopoietic system, especially HSCs, responds to CR is key to enhancing the positive effects of CR while minimizing immune suppression, a previously reported phenotype[7–9]. In this study, we found that male mice

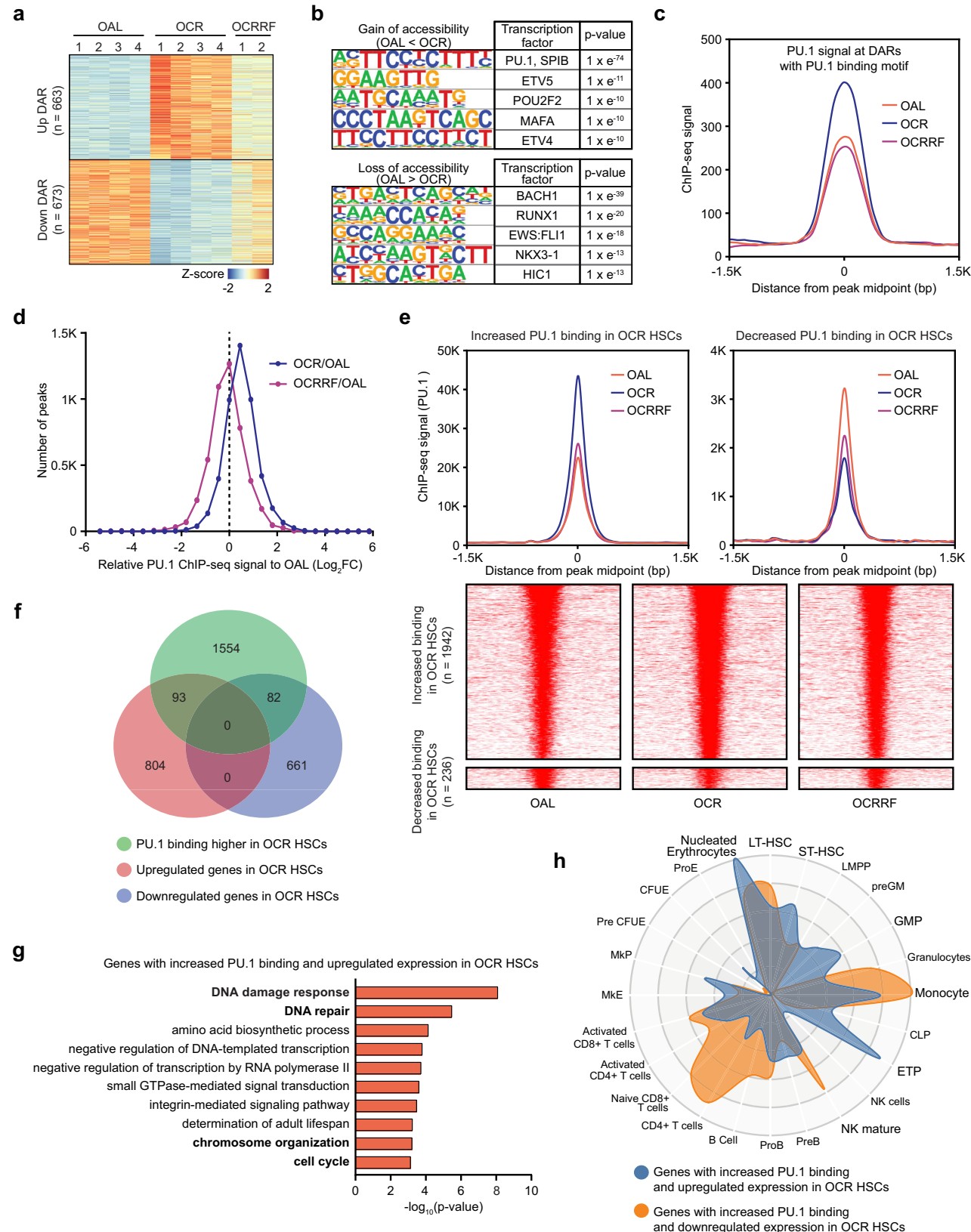

respond to this level of CR by reducing total white blood cell production, though whole bone marrow cellularity was not significantly decreased. Notably, while both myeloid and lymphoid cell production were reduced, myeloid cells - especially those critical for survival, such as MEPs, granulocytes, platelets, and red blood cells−were preferentially maintained. In contrast, lymphoid cell production was primarily suppressed, from early progenitor cells (MPPFlk2+ and CLPs) to differentiated B220+ blood cells. These findings are similar to the phenotypes seen after one year of 30% CR reported by the Rudolf group[9]. Unexpectedly, we observed increased cycling of HSCs under 40% CR conditions, which did not lead to enhanced self-renewal but rather promoted differentiation into myeloid cells. Although lymphoid

**Fig. 5 | PU.1 regulates the response of HSCs to CR by increased binding to its target genes. a** Heatmap of differentially accessible regions (DARs) between OAL and OCR HSCs (FC > 1.5, FDR < 0.05). **b** Transcription factors (TF) with binding motifs enriched in regions exhibiting CR-induced changes in accessibility. **c** Composite plot of PU.1 ChIP-seq signals at DARs containing PU.1 binding motif. **d** Comparison of PU.1 ChIP-seq signals at all peaks in OCR and OCRRF HSCs relative to OAL HSCs. **e** Heatmap and composite plots of PU.1 ChIP-seq signals generated from peaks with altered PU.1 binding (FC > 1.5) in OAL versus OCR HSCs comparison. **f** Venn diagram showing the overlap between genes with increased PU.1 binding and DEGs from OAL versus OCR comparison. **g** Pathway analysis of genes with increased PU.1 binding and upregulated expression in OCR HSCs. Gene Ontology enrichment was performed using DAVID. Enrichment *p*-values correspond to the DAVID EASE Score (modified one-sided Fisher's exact test) without correction for multiple testing. **h** CellRadar plot derived from the overlapping genes between those with increased PU.1 binding and the DEGs in the OAL versus OCR comparison.

differentiation was suppressed during CR, we observed a burst in lymphoid cell production upon refeeding, sometimes exceeding levels seen in ad lib fed mice. This suggests that modified feeding paradigms, such as cyclic CR or intermittent fasting, may preserve the benefits of CR while potentially maintaining or even enhancing immune function[23]. In terms of human health and disease prevention, intermittent or periodic CR may also be more practical and sustainable compared to lifelong CR.

The gold standard assays to test for functional potential, transplant assays, have previously shown lifelong CR does not affect HSCs' function[14,15], we also report no significant alterations in this paradigm (Supplementary Fig. 9), and we propose lack of functional translation of the altered HSCs' transcriptional profiles is due to the swift reversal of CR-induced phenotypes when exposed to a fed environment. After refeeding, CR-associated changes in PB parameters, bone marrow cell frequencies, and HSC transcriptomes returned to levels similar to those seen in ad lib fed mice after just four weeks, and transplant experiments are four months. As recipient mice in transplantation assays were not CR treated - effectively creating a refed environment for the donor CR HSCs - this may contribute to the observed lack of differences between CR and control HSCs.

Importantly, our results showed that lifelong CR affects the HSC compartment transcriptionally, and the increased cycling did not appear to accelerate aging phenotypes; instead, a more youthful transcriptional state was present in lifelong CR HSCs. We examined whether other notable benefits associated with CR reported in other adult stem cells, including increased autophagy[24,25] and changes in mTORC and Sirt1 pathways[26,27] occurred in HSCs, but there were no significant enrichments for these pathways or targets. While most transcripts involved in these processes were not significantly altered, we did see a significant decrease in *Prkaa2*, *Ulk1*, and *Sqstm1 (p62)* (Supplementary Data 1) in the OCR HSCs compared to OAL, but didn't observe these differences in the YCR. While in somatic cells, CR has been reported to show metabolic remodeling, in muscle stem cells, no changes in metabolism were seen in calorically restricted animals[28]. While we did not directly measure changes in mitochondria metabolism, under CR conditions the old HSCs did show a decrease in both *Cpt1a* and *Acadl* which suggest potential reduction in fatty acid metabolism, which could affect the HSCs potential[29], but these transcripts revert back to levels seen under ad lib feeding conditions which may explain the lack of this phenotype in the transplantation setting.

To better understand how CR mitigated acquiring an aged transcriptional signature, we evaluated associated epigenetic alterations. Surprisingly, little information about histone methylation under caloric restriction is available, with the majority of studies focused on histone acetylation (due to the SIRT1 connection), and the majority of studies performed in yeast or invertebrate models. We find globally a reduction in both H3K4me3 and H3K27me3 in the OCR HSCs compared to OAL. This is similar to changes that occur in liver cells under methionine restriction[30], but in contrast to brains of female BALB/c with 30% caloric restriction, which showed increase in global H3K27me3. The global decrease in both marks in the OCR compared to OAL was striking to us as during aging we also see loss of these histone modifications in old HCSs compared to young (manuscript under review). Given that the transcriptome of the OCR HSCs was younger,

we may have expected to see increased levels of both. Then, we examined loci that had epigenetic and corresponding transcriptional changes and focused on those with relatively increased levels of H3H27me3 and decreased gene expression. This profiling identified several receptors from the HSC aging signature gene list that show increased expression in aged HSCs[17] but decreased expression in the OCR HSCs. We focused on *Kdr* and *Bmpr1a* as they showed the most robust relative increase of H3K27me3, associated with decreased expression of these transcripts in OCR HSCs.

Knockdown of *Kdr* and *Bmpr1a* in aged HSCs isolated from ad lib fed mice improved their aging transcriptome, suggesting that these receptors' related pathways are critical in shaping HSC aging characteristics. We designed our transplant paradigm to evaluate the effects of sustained knockdown on transcription in old HSCs, but by prioritizing this aspect, we were precluded from analyzing the functional output of the aged HSCs. Since we minimized the time of exposure to the viruses ex-vivo (up to 24 h) and didn't sort for GFP or mCherry expression before transplantation, we are unable to determine how competitive each of the transplants were, as we transplanted both infected and non-infected HSCs precluding analysis of changes in functional potential. However, we were able to demonstrate that these key receptors play an important role in regulating the HSC transcriptome.

We believe that KDR will be an intriguing receptor to modulate to mitigate aging HSCs phenotypes as KDR is a vascular endothelial growth factor receptor crucial for angiogenesis, vascular development and HSC survival[31,32]. VEGF signaling has been implicated in aging phenotypes[33], and in tumor models, targeting KDR restored frequencies of HSC populations. KDR has also been implicated in mediating differentiation choices in inflamed cancer conditions, with KDR inhibiting lymphoid cell development[34]. Intriguingly, PU.1 is also essential for differentiation choices, involved in both myeloid and lymphoid differentiation[35,36]. Further PU.1 has been shown to modulate the HSC cell cycle, either preventing HSC exhaustion or limiting their expansion during inflammatory stress, primarily acting as a repressor of HSC cycling[21,37]; however, under CR conditions PU.1 acts as an activator of HSC cycling, indicating a context-dependent role. We have recently shown that during aging, changes in the chromatin accessibility of aged HSCs include the loss of accessibility at PU.1 binding sites, and increased access to RUNX1 and EWS:FLI1 binding sites (manuscript under review), all of which were significantly reversed in the lifelong CR HSCs.

We have presented data suggesting that HSCs have a unique response to CR and again highlight how individual cell types respond to 40% CR in diverse ways. HSCs increase their cycling and prioritize for generating life-critical cell types. We also demonstrate that methylation at H3K4 and H3K27 was globally reduced in OCR HSCs compared to OAL HSCs and that these histone methylation modifications respond to dietary interventions. Further, we identify two key receptors, that significantly increase with age on HSCs, and reducing their expression levels restores transcription to a more youthful signature. These receptors act in concert with changes in PU.1 binding under CR conditions mitigating age associated alterations in transcription. These data presented will inform future studies on drivers of the positive aspects of CR on hematopoiesis and suggest different cell

types will respond to CR in distinct manners depending on their function and that inclusion of additional epigenetic modifications should be evaluated in dietary restriction studies.

It should be noted that this study was conducted exclusively in male mice, and future work will be necessary to determine whether similar mechanisms are conserved across sexes. Additionally, while we did not perform direct functional validation of immune suppression, previous studies indicating this response were validated by our results of suppressed lymphoid cell production, from early progenitor cells (MPPFlk2+ and CLPs) to differentiated B220+ blood cells at 40% CR.

## Methods

### Animals
Life-long ad lib feeding and CR mice (male, C57BL/6, 24-26 months) were acquired from the NIA Aged Rodent Colony. Young C57BL/6 (CD45.2) male mice (3-4 months), Ki67-RFP reporter mice, and young female transplant recipient B6.SJL-Ptprca Pepcb/BoyJ (CD45.1) mice were obtained from The Jackson Laboratory. Lifelong CR mice were subjected to 40% CR (60% of AL diet) starting at the age of 3-4 months. Short-term CR mice (3-4 months) were subjected to 20% CR for the first week, then 40% CR for another 3 weeks (C57BL/6) or 3 months (Ki67-RFP reporter mice). For refeeding, CR mice were returned to ad lib feeding for 4 weeks following either lifelong or short-term CR. All CR regimens were applied continuously, with the same restricted feeding maintained on both weekdays and weekends. Mice were maintained under a 12 h light/12 h dark cycle, at an ambient temperature of 69–75 °F, with relative humidity of 30–70%. All animal experiments were performed under protocols approved by NIA Institutional Animal Care and Use Committees (469-TGB-2025).

### Antibodies
**Flow cytometry.** The following antibodies are from Biolegend: Biotin anti-TER119 (Cat# 116204, 1:100 dilution), Biotin anti-B220 (Cat# 103204, 1:100 dilution), Biotin anti-CD3 (Cat# 100244, 1:100 dilution), Biotin anti-Mac-1 (Cat# 101204, 1:100 dilution), Biotin anti-Gr-1 (Cat# 108404, 1:100 dilution), PB anti-TER119 (Cat# 116232, 1:200 dilution), PB anti-B220 (Cat# 103227, 1:200 dilution), PB anti-Mac-1 (Cat# 101224, 1:200 dilution), PB anti-CD3 (Cat# 100214, 1:200 dilution), BV421 anti-Gr-1 (Cat# 108445, 1:200 dilution), BV421 anti-IL7Rα (Cat# 135027, 1:200 dilution), APC/Cy7 anti-Sca-1 (Cat# 108126, 1:200 dilution), PE anti-c-Kit (Cat# 105808, 1:200 dilution), BV421 anti-c-Kit (Cat# 105827, 1:200 dilution), APC anti-Flk2 (Cat# 135310, 1:50 dilution), PE/Cy7 anti-CD150 (Cat# 115914, 1:200 dilution), APC anti-CD48 (Cat# 103412, 1:200 dilution), APC anti-CD45.1 (Cat# 110714, 1:200 dilution), PerCP/Cy5.5 anti-CD45.1 (Cat# 110728, 1:100 dilution), PB anti-CD45.2 (Cat# 109820, 1:100 dilution), PerCP/Cy5.5 anti-TER119 (Cat# 116228, 1:200 dilution), APC/Cy7 anti-B220 (Cat# 103224, 1:200 dilution), PE anti-CD3 (Cat# 100206, 1:200 dilution), PE/Cy7 anti-Mac-1 (Cat# 101216, 1:200 dilution), FITC anti-Gr-1 (Cat# 108406, 1:200 dilution), BV510 anti-Gr-1 (Cat# 108457, 1:200 dilution). FITC anti-CD34 (Cat# 11-0341-85, 1:50 dilution) and PerCP/Cy5.5 anti-FcγRα (Cat# 45-0161-82, 1:100 dilution) are from ThermoFischer.

**ChIP-seq.** Anti-H3K4me3 (Sigma, Cat# 07-473, 1.5 ul per IP), anti-H3K27me3 (Diagenode, Cat# C15410195, 1.5 ul per IP), anti- PU.1 (ThermoFischer, Cat# MA5-15064, 9 ul per IP), anti- PU.1 (abcam, Cat# ab227835, 9 ul per IP).

### Peripheral blood analysis
Complete blood count (CBC) data were collected using Hemavet 950FS. Frequencies of PB cell populations were generated using flow cytometry after ACK treatment and staining with Ter119, B220, Mac-1, CD3, and Gr-1 for CR experiments; staining with CD45.1, CD45.2, Ter119, B220, Mac-1, CD3, and Gr-1 for transplantation experiments.

### Whole bone marrow analysis
Bone marrow cells were treated with ACK and then stained with antibodies. For staining WBM cells from ad lib and CR mice: lineage (Ter119, B220, Mac-1, CD3, Gr-1), Sca-1, c-Kit, CD34, Flk2, CD150, FcγRα. For staining WBM cells from post-transplant recipient mice: lineage (Ter119, B220, Mac-1, CD3, Gr-1), Sca-1, c-Kit, CD34, Flk2, CD150, CD45.1, CD45.2.

### HSC purification
Bone marrow cells were collected from crushed femurs, tibias, pelvises, humeri and filtered through 100 μm filters. Isolated BM cells were positively selected with PE c-Kit antibody and EasySep™ PE Positive Selection Kit II (17684, STEMCELL). For CR experiments, enriched cells were stained with antibodies against lineage (Ter119, B220, Mac-1, CD3, Gr-1, IL-7Rα), Sca-1, c-Kit, CD34, Flk2, CD150. For transplantation experiments, enriched cells were stained with antibodies against lineage (Ter119, B220, Mac-1, CD3, Gr-1), Sca-1, c-Kit, CD48, CD150, CD45.1, CD45.2. Propidium Iodide (PI) was used to exclude dead cells. HSCs were sorted as PI⁻Lin⁻Sca-1⁺c-Kit⁺(LSK) CD34⁻Flk2⁻CD150⁺ for CR experiments and PI⁻LSKCD48⁻CD150⁺CD45.2⁺ for transplantation experiments with BD FACSAria II Cell Sorter.

### HSC in vitro culture and virus transduction
HSCs were cultured in media described before[38]: F12 medium, 10 mM HEPES, 1% ITSX, 1% P/S/G, 100 ng/ml mouse TPO, 10 ng/ml mouse SCF and 0.1% PVA (P8136, Sigma) in 96-well U-bottom plates at 37 °C with 5% $CO_2$ and $O_2$. The lentivirus shRNA knockdown vector construction and virus packaging were performed by VectorBuilder. The target sequences of shRNA are 5'-GCATCTCATCTGTTACAGCTT-3' (Kdr-shRNA5), 5'-CAATTTGTGCAACCAGTA-TTT-3' (Bmpr1a-shRNA1), and 5'-TCAAGACTCCAATCCTGATAA-3' (Bmpr1a-shRNA2). For virus transduction, 10,000 HSCs purified from aged mice were cultured in 50 μl of media with the virus overnight (12–16 h). The next morning, 150 μl of fresh media was added to each well. For knockdown efficiency test, HSCs were cultured for 3.5 days before sorting for RNA-seq. For transplantation, HSCs were cultured for 20–24 h before the procedure.

### HSC transplantation
For transplantation of HSCs from OAL, OCR and OCRRF mice, 200 HSCs (CD45.2) were transplanted into lethally irradiated (9.56 Gy) recipients (CD45.1) together with $2 \times 10^5$ WBM cells (CD45.1) via retro-orbital injection. For transplantation of virus transfected HSCs, about 500 cultured HSCs (CD45.2) were transplanted into lethally irradiated (9.56 Gy) recipients (CD45.1) together with 30,000 myeloid progenitor cells (CD45.1) via retro-orbital injection.

### RNA-seq
HSCs were directly sorted into TRIzol Reagent (15596026, Thermo-Fisher) and stored at −80 °C. RNA was purified with Direct-zol RNA Microprep (R2060, Zymo) for CR experiments. RNA from transplanted HSCs was extracted using phenol-chloroform extraction and iso-propanol precipitation. cDNA libraries were prepared with SMART-Seq RNA Kit (634891 for CR experiments, 634772 for RNA from transplanted HSCs, TaKaRa) according to the manufacturer's protocol. Sequencing libraries were constructed using Nextera XT DNA Library Preparation Kit (FC-131-1024, Illumina) with 125 pg input cDNA. Sequencing was done with Illumina HiSeq 2500 instrument for CR experiments libraries, and Illumina NovaSeq 6000 System for libraries from transplanted HSCs.

### ATAC-seq
ATAC-seq was performed as described previously[39]. 6,000 sorted HSCs were spun down using a fixed angle rotor at 500 g, 4 °C for 10 min. Cells were resuspended with 50 μl of cold resuspension buffer (10 mM Tris-HCl, 10 mM NaCl, 3 mM $MgCl_2$ in water) containing 0.1%

NP40, 0.1% Tween-20, and 0.01% Digitonin by pipetting up and down 3 times, and incubated on ice for 3 min. 1 ml of cold resuspension buffer containing 0.1% Tween-20 (without NP40 or Digitonin) was added to wash the cells and tubes were inverted 3 times to mix. The nuclei were pelleted by spinning at 500 g, 4 °C for 10 min with a fixed angle rotor. The pellet was then resuspended in 50 μl of transposition mix (25 μl of 2x TD buffer, 2.5 μl of Tn5 Transposase (20034197, Illumina), 22.5 μl of $H_2O$) by pipetting up and down 6 times. The reaction was Incubated at 37 °C for 30 min in a thermomixer with 1000 rpm mixing and cleanup was performed with DNA Clean & Concentrator-5 (D4013, Zymo Research). DNA was eluted with 21 μl of $H_2O$ and amplified for 5 cycles with NEBNext Ultra II Q5 Master Mix (M0544L, NEB). We determined the optimum additional PCR cycles by qPCR as described previously[40]. The final PCR reaction was purified with DNA Clean & Concentrator-5 (D4013, Zymo Research). Libraries were size selected (120 – 800 bp) by running TBE gel (EC6265BOX, ThermoFisher) and sequenced on Illumina HiSeq 2500 instrument.

## ChIP-seq

ChIP-seq was performed as described previously[41] with Chromatin Immunoprecipitation (ChIP) Assay Kit (17-295, Millipore). For histone ChIP-seq, 10,000 sorted HSCs were mixed with $1 \times 10^8$ bacteria (18258012, ThermoFisher), then crosslinked with 1% formaldehyde (F8775, Sigma-Aldrich) at room temperature (RT) for 10 min. Cross-linking was stopped by adding 2 M glycine to a final concentration of 0.125 mM and incubated at RT for 5 min. After one wash with cold 1x PBS, cells were resuspended in SDS lysis buffer with 1x proteinase inhibitor cocktail (Millipore, 4693132001) and incubated on ice for 10 min. Sonication was performed with Bioruptor Plus (Diagnode) using the following setting: 3 × 10 cycles of 30 sec ON and 30 sec OFF at HIGH setting. After sonication, fragmented chromatin was diluted by adding 9 volumes of ChIP dilution buffer with 1x proteinase inhibitor cocktail. Corresponding antibodies were added and incubated overnight at 4 °C with rotation. 20 μl of Dynabeads Protein A (10002D, ThermoFisher) were added and incubated at 4 °C for 2 h with rotation. Before washing, 5 ng biotin-DNA carrier immobilized with 10 μl of Dynabeads M-280 Streptavidin (11205D, ThermoFisher) was added. Sample was washed with 1 ml washing buffer at 4 °C for 5 min in the following order: 1x with low salt buffer, 1x with high salt buffer, 1x with LiCl washing buffer, 2x with TE. Beads were resuspended with decrosslinking buffer (0.5% SDS, 0.2 M NaCl in TE) and incubated at 65 °C for 4–5 hour or overnight with 1000 rpm mixing. 20 mg/ml proteinase K (final concentration 0.8 mg/ml, 26160, ThermoFisher) was added and incubated at 55 °C for 2 h. ChIPed DNA was purified with 1.6x SPRIselect beads (B23318, Beckman Colter). For PU.1 ChIP-seq, 20,000 sorted HSCs were mixed with $5 \times 10^7$ bacteria, and beads were washed for 1 min instead of 5 min in the sample wash step. Libraries were prepared with TruSeq Nano DNA Low Throughput Library Prep Kit (20015964, Illumina) according to manufacturer's instructions. Sequencing was done with Illumina HiSeq 2500 instrument.

## RNA-seq analysis

For analysis of transcriptome datasets, we built an index for STAR using the GENCODE M22 reference feature including protein-coding and non-coding genes. Prior to sequence alignment, we applied trim galore (version 0.4.3) with cutadapt (version 1.12)[42] to remove any unnecessary genomic fragments (e.g., adapter dimers) and low-quality nucleotide sequences from the raw reads. We mapped adapter trimmed sequencing reads to the mouse reference genome (mm10) using STAR aligner[43] and calculated the raw count using featureCounts software (gene-level)[44]. Differentially expressed gene (DEG) lists were generated with DESeq2[45]. For transposable element (TE) detection, we utilized SQuIRE[46] and used limma based edgeR package of R[47] to find differential transposable elements.

## ATAC-seq analysis

All sequencing reads were trimmed using cutadapt[42], and trimmed reads (>36 bp minimum alignment length) were mapped against the mm10 genome using BWA aligner[48]. We used de-duplicated and uniquely mapped reads for peak calling analysis after excluding high-sensitive black-list regions defined by ENCODE. The candidate peaks were predicted by MACS peak calling tool (FDR < 0.01)[49]. After identifying narrow peaks from all replicates, we created a merged set of consensus peaks and generated a matrix of open chromatin regions (OCRs). This OCR matrix was then imported into the R package DESeq2[45], and we determined differentially accessible regions (DARs) with cutoff: FC > 1.5, logCPM > 1.5, FDR < 0.05. Finally, the candidate differential open chromatin regions were submitted to search for potential transcription factor binding sites using HOMER software[50] with non-DARs as background regions. In this analysis, de novo motif and known motif searches were performed, and we reported the top five significant de novo motif results. Composite plots and heatmaps were generated by customized Python script and Java TreeView software.

## ChIP-seq analysis

All sequencing reads were trimmed using cutadapt[42], and trimmed reads (>36 bp minimum alignment length) were mapped against the mm10 genome using BWA aligner[48]. We used de-duplicated and uniquely mapped reads for peak calling analysis after excluding high-sensitive black-list regions defined by ENCODE. For H3K4me3 ChIP-seq, signals within ±1 kb of the transcription start site (TSS) were used for downstream analysis. For H3K27me3 ChIP-seq, signals from 0 to 2 kb downstream of the TSS were used for further analysis. For PU.1 ChIP-seq, the candidate peaks were predicted by MACS peak calling tool (FDR < 0.05)[49]. Composite plots and heatmaps were generated with customized Python script and Java TreeView software.

## Reporting summary

Further information on research design is available in the Nature Portfolio Reporting Summary linked to this article.

## Data availability

The raw data generated in this study have been deposited in the NCBI Gene Expression Omnibus database under accession code GSE284988. Source data are provided with this paper.

## Code availability

All customized Python scripts used in the manuscript are available via GitHub repository URL (https://github.com/genomicspark/ESCA_Unit_Scripts) and has been archived in Zenodo for citation (https://doi.org/10.5281/zenodo.17857712)[51]. Source data are provided with this paper.

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

## Acknowledgements

Many thanks to Drs. Rafael de Cabo, Myriam Gorospe and all members of TGB for invaluable support. We thank Dr. Fei Ma for assistance in preparing illustrations used in this manuscript, including generating and assembling figure elements created with BioRender. Kind support was shared by the Genomics Core and we'd like to specially acknowledge Jinshui Fan, William Wood, and Supriyo De for their assistance with data handling and sequencing advice. Thanks to Christopher Dunn at the NIA Flow Core for always being helpful and willing to share time and expertise. We would like to thank all the members of the NIA Comparative Medicine Section for their consistent efforts and high standards of animal care. Data analysis of this work utilized the computational resources of the NIH HPC Biowulf cluster (http://hpc.nih.gov). This research was supported entirely by the Intramural Research Program of the NIH, National Institute on Aging.

## Author contributions

L.Z., F.T., W.K., M.T., and K.L. performed the experiments; B.P. performed bioinformatical analyses; L.Z. and I.B. designed the research, interpreted the results, and wrote the manuscript.

## Funding

## Competing interests

The authors declare no competing interests.
