## [Transparent Peer Review file · Nature Communications]

Epigenetic profiling of hematopoietic stem cells from male mice identifies KDR and PU.1 as regulators of aging transcriptome and caloric restriction response

Corresponding Author: Dr Isabel Beerman

Version 0:

Reviewer comments:

Reviewer #1

(Remarks to the Author)

Comments for the authors:

The manuscript by Dr. Zong examines mechanisms underlying HSC responses to CR which lead to loss of immune function. They use both life-long and short-term CR in mouse models and found a decrease in WBC counts with preservation in myeloid-biased and megakaryocyte/platelet-erythroid biased progeny. They conclude that this CR response prioritizes red cells, platelets, and innate immune cells at the expense of adaptive immune cell differentiation. HSC under CR are driven into cell cycle to support myeloid bias over self-renewal. Intriguingly, they also find that CR mitigates aging-associated transcriptomic changes despite the myeloid and megakaryocyte/platelet-erythroid biases. These transcriptomic changes are lost after re-feeding ad lib. Using epigenetic profiling, they found that KDR is a central regulator of the CR response and knock-down of Kdr in aged, murine HSCs enforces the youthful transcriptome that was observed in lifelong CR. Using ATACseq, they also identify PU.1 as a regulator of the CR response by binding to downstream target genes. Overall, this study uncovers exciting mechanistic insight relevant to CR and HSC with aging, although some additional analyses that provide insight into how these changes are linked would substantiate the results and strengthen the conclusions.

General comments:

The research focus is of great interest in the field because it unveils previously unknown hematologic responses to CR and aging. Additional bioinformatic analyses to establish a potential link between KDR and PU.1 would strengthen the manuscript. Further examination of the long-term HSC pool with OACR would also improve the paper since it is not clear how CR promotes a decrease in LT-HSC together with HSC cycling and myeloid bias while also repressing Aging Signature (AS) genes; perhaps listing the specific AS genes in a Table would be illuminating. Examination of quiescent (CD41 low/Procr high) HSC compared to cycling (CD41 high/Procr low) HSC could help to further define changes in HSC subsets with CR since the finding that HSC cycling and myeloid bias is difficult to understand in the context of mitigating aging transcriptional changes unless quiescent HSC are maintained while the remaining HSC that cycle are biased toward myeloid lineages. Once these issues are addressed, this interesting work would be suitable for publication.

Major issues:

Figure 1:

Fig 1f-g: The text in paragraph 1 of results states that RBC and platelet numbers are maintained under CR conditions and Fig 1f-g show no significant difference between OAL and CR. Are RBC and platelet numbers similar to those observed in young mice? (A bar graph in the main figure would be helpful here to avoid searching for comparisons in the supplemental data).

Fig 1: Could you also add comparisons between OAL and OCRRF? (OAL vs. OCR and OCR vs. OCRRF are shown, presumably OAL vs. OCRRF are similar, but please show for clarity)

Figure 2:

Please provide details in text for HSC markers used in RNAseq so readers do not need to search through methods.

Fig 2d: It may be helpful to perform a GSEA comparing OCR (on left) versus OAL (on right) which would yield positive (+) NES (normalized enrichment scores) for cell cycling pathways, and better match the text and Fig. 1c stating that OCR activates cell cycle progression genes. The data is convincing; it would be easier to follow if the enrichment scores match the text and conclusions.

Fig. 2g: OCR seems to expand preB –is this significant? If yes, could this be discussed and investigated further? Is there a block in differentiation since B cells are down-regulated?

Extended Figure 4:

The trend noted with more down-regulated genes in YCR versus YAL seems significant from the heat maps (Extended Fig 4) – are there other statistical approaches to compare the proportion of up versus down-regulated genes?

Figure 3:

Fig. 3a: AS dysregulated genes are shown in a bar graph, but not the specific genes. A table/excel file of these genes and their differential expression and P values would be of interest.

Please discuss the genes *Clu*, *Osmr*, *Ect2*, *Mcm5* further since they were discussed in the text and examined in Fig. 3e, although their full gene names and function may not be well-known to many readers.

Figure 4:

The text mentions that sirtuin genes were not altered with CR, although genes regulating histone methylation were reported to be differentially regulated. Please elaborate further and show the genes and data supporting this.

Please also provide rationale for focusing on H3K4me3 and H3K427me3. Did the gene expression results suggest these marks would be affected?

Fig. 4c-i: Gene silencing experiments suggest that KDR is a key regulator of the CR response and effects on aging signature.

Text Line 200 – Please clarify: Alternatively, we could have examined the 7 genes with increased H3K4me3 that mitigated the decreased expression associated with aging in HSC. I would either examine these genes or not mention that you could examine them.

Figure 5:

Based on ATACseq, PU.1 is implicated as a primary regulator of the CR response which was validated by ChIPseq. What is the link between KDR signaling and PU.1? Could experiments or bioinformatic analyses be performed to identify potential pathways that link these 2 factors?

Some additional issues to address:

The findings that CR increases HSC cycling with a myeloid bias while mitigating AS genes seems contradictory. How does the LT-HSC in OCR compare to Young – is CR restraining HSC expansion that occurs with aging?

Along these lines, are the expanded HSC acquiring mutations as occurs with human HSC aging? Although this is beyond the scope of this paper, could the authors speculate on whether CR may modulate acquisition of mutations that allow HSC expansion?

How can we reconcile increased HSC cycling and myeloid bias (2 aging phenotypes) with dampening of the AS? What are the specific AS genes?

Are the quiescent LT-HSC being preserved with CR? Examination of quiescent (CD41 low/*Procr* high and/or other markers of quiescence) versus cycling LT-HSC (CD41 high/*Procr* low and/or other markers of cycling) could provide further insight into how CR represses AS while promoting cycling and myeloid bias.

While beyond the scope of this paper, transcriptomic analyses of spleens and spleen/bone marrow single cell analyses would provide further insight into mechanisms underlying CR with aging.

Reviewer #2

(Remarks to the Author)

Summary

This manuscript provides a mechanistically detailed and well-executed investigation into how caloric restriction (CR) remodels the aging hematopoietic stem cell (HSC) compartment. The authors report that CR promotes a myeloid-biased transcriptional program at the expense of lymphoid differentiation, driven in part by epigenetic silencing of *Kdr* and *Bmpr1a* and increased PU.1 chromatin occupancy. The findings are timely and impactful; however, several areas require further clarification or additional data to strengthen the manuscript's claims and contextualization.

Specific suggestions:

Definition of "Youthfulness"

While the manuscript asserts that CR preserves a youthful HSC transcriptome, the observed phenotype is also

characterized by reduced lymphoid output—particularly naive T cells—which are hallmarks of youth. The current phenotype may instead reflect a regulated or functionally adaptive response to energetic stress. I recommend refining the language throughout to differentiate between transcriptional rejuvenation and functional youthfulness. A deeper analysis of which aged HSC genes are reversed by CR and which persist would clarify the extent of this rejuvenation.

Direct comparison across age and CR groups

Although the authors refer to a youthful signature, they do not show a PCA or clustering analysis that includes both young ad libitum (YAL) and young CR (YCR) HSCs alongside old CR (OCR) HSCs. Including these groups in PCA or trajectory analyses would allow the readers to evaluate whether OCR HSCs genuinely converge toward a youthful transcriptional profile or instead form a unique CR-adapted state.

Clarification of CR protocol

More detail is needed on the CR paradigm, particularly concerning the weekend feeding regimen. Were mice subjected to continuous daily restriction, or did they receive multi-day rations? Such patterns could lead to unintentional refeeding-fasting cycles, influencing immune and HSC outcomes. Clarifying whether CR was continuous or intermittent is important for interpreting discrepancies with previous reports and for translational relevance.

T cell output

The authors quantify CD3⁺ cells in their gating strategy, yet T cell data are not reported. This omission is notable, as it precludes comparison with studies (e.g., Di Francesco et al. Nature 2024) that observed CR-mediated preservation of T cells. Reporting absolute and relative T cell numbers would help resolve whether CR compromises adaptive immunity broadly or selectively suppresses B cells.

Functional evidence for immune suppression

The claim that CR may compromise immune function would be strengthened by functional validation, such as infection or tumor challenge models. In the absence of these experiments, the authors should clearly acknowledge this limitation and revise language in the abstract and discussion to avoid overinterpretation.

Biological replicates in transcriptomic analyses

In Figure 2a, only n=2 samples are included for the OCRRF group, while other groups have n=4. The authors should explain why OCRRF replicates were reduced and clarify whether conclusions drawn from this group are robust. If possible, additional replicates should be included.

Bone marrow cellularity and absolute counts

The manuscript reports frequencies of HSCs and progenitor populations but omits data on total BM cellularity. Without absolute counts, it is difficult to distinguish between genuine expansion/contraction of cell populations and shifts due to altered gating proportions. Including BM cellularity and absolute numbers of cell populations would substantially strengthen the phenotypic conclusions.

Kdr ligand expression

The authors should evaluate whether expression of Kdr ligands (e.g., VEGFs) is altered with aging or CR, either in their dataset or via comparison with public datasets. This would strengthen the case for Kdr as a physiologically relevant modulator of HSC aging.

PU.1 and Kdr Pathway Integration

The manuscript presents parallel evidence for PU.1 and Kdr as CR response regulators but does not clarify whether these pathways intersect mechanistically. Is there evidence that PU.1 binding is altered at loci with CR-induced histone changes? Are any PU.1-bound genes among those regulated by Kdr? Exploring overlap in differential chromatin accessibility and histone methylation could uncover a more unified mechanism.

PU.1 Binding Partners

For genes with differential PU.1 occupancy, are different adjacent transcription factor motifs enriched among upregulated vs downregulated genes? This could suggest context-specific co-factors that shape PU.1's dual roles as an activator and repressor. This analysis would provide important mechanistic depth.

Minor points

Currently the text mentions B220⁺ cells (line 93). Is B220 used as a marker for B cells? If so, the text should state that there is decrease in B cells with both short-term and long term CR for clarification.

OWT is used in the text starting on line 258. This most likely refers to the Old Ad Lib (OAL) mice and should be corrected for consistency.

Reviewer #3

(Remarks to the Author)

In this manuscript, Zong et al. investigate the mechanisms underlying the effects of caloric restriction (CR) on the immune function of aged mice. They first confirm that CR leads to reduced white blood cell counts and myeloid bias. They then examined the effects of short term and long term CR as well as refeeding on transcription aging signatures in hematopoietic stem cells (HSCs) and demonstrate that CR preserves the 'youthful' signature. They examined epigenetic changes in specific histone methylation marks which identified Kdr and Bmpr1a as candidate regulators of the CR effects. Subsequent knockdown experiments confirmed that Kdr is able to recapitulate the CR effect. Examination of differential open chromatin regions further implicated Pu.1 as a candidate inhibitor of HSC renewal.

This work substantially advances our understanding the molecular mechanisms underlying immune response to CR.

The experiments are clearly described and follow a logical progression that supports the conclusions.

My only major concern is that the study is limited to male C57BL/6J mice. Including female mice should be considered essential, despite previous studies reporting absence of CR effects. Further the use of a single genetic background limits the scope of the conclusions as it is unclear if these responses are universal or limited to (male) C57 mice. Nonetheless the study is valuable and advances new hypotheses that can be explored in other contexts.

Reviewer #4

(Remarks to the Author)

Version 1:

Reviewer comments:

Reviewer #1

(Remarks to the Author)

The authors have addressed the critiques and the revised manuscript is appropriate for publication.

Reviewer #2

(Remarks to the Author)

I appreciate the authors' thorough revision of the manuscript and detailed responses to the previous comments. The manuscript has substantially improved and the new analyses and clarifications address the questions raised previously. In my opinion, the manuscript is suitable for publication in Nature Communications.

Reviewer #3

(Remarks to the Author)

No additional concerns to note.

REVIEWER COMMENTS

Reviewer #1 (Remarks to the Author):

Comments for the authors:

The manuscript by Dr. Zong examines mechanisms underlying HSC responses to CR which lead to loss of immune function. They use both life-long and short-term CR in mouse models and found a decrease in WBC counts with preservation in myeloid-biased and megakaryocyte/platelet-erythroid biased progeny. They conclude that this CR response prioritizes red cells, platelets, and innate immune cells at the expense of adaptive immune cell differentiation. HSC under CR are driven into cell cycle to support myeloid bias over self-renewal. Intriguingly, they also find that CR mitigates aging-associated transcriptomic changes despite the myeloid and megakaryocyte/platelet-erythroid biases. These transcriptomic changes are lost after re-feeding ad lib. Using epigenetic profiling, they found that KDR is a central regulator of the CR response and knock-down of Kdr in aged, murine HSCs enforces the youthful transcriptome that was observed in lifelong CR. Using ATACseq, they also identify PU.1 as a regulator of the CR response by binding to downstream target genes. Overall, this study uncovers exciting mechanistic insight relevant to CR and HSC with aging, although some additional analyses that provide insight into how these changes are linked would substantiate the results and strengthen the conclusions.

General comments:

The research focus is of great interest in the field because it unveils previously unknown hematologic responses to CR and aging. Additional bioinformatic analyses to establish a potential link between KDR and PU.1 would strengthen the manuscript. Further examination of the long-term HSC pool with OACR would also improve the paper since it is not clear how CR promotes a decrease in LT-HSC together with HSC cycling and myeloid bias while also repressing Aging Signature (AS) genes; perhaps listing the specific AS genes in a Table would be illuminating. Examination of quiescent (CD41 low/Procr high) HSC compared to cycling (CD41 high/Procr low) HSC could help to further define changes in HSC subsets with CR since the finding that HSC cycling and myeloid bias is difficult to understand in the context of mitigating aging transcriptional changes unless quiescent HSC are maintained while the remaining HSC that cycle are biased toward myeloid lineages. Once these issues are addressed, this interesting work would be suitable for publication.

Response: We thank the reviewer for the positive assessment of our work and for the thoughtful and constructive comments. The questions and suggestions raised are highly valuable and have helped us further improve the clarity and depth of our study. We have addressed all these points in detail in the following point-by-point responses.

Major issues:

Figure 1:

Fig 1f-g: The text in paragraph 1 of results states that RBC and platelet numbers are maintained under CR conditions and Fig 1f-g show no significant difference between OAL and CR. Are RBC and platelet numbers similar to those observed in young mice? (A bar graph in the main figure would be helpful here to avoid searching for comparisons in the supplemental data).

Response: We thank the reviewer for this helpful comment. Compared to young C57BL/6J male mice, the number of RBCs is decreased (Fig. 1f), while the number of platelets is increased in aged mice (Fig. 1g). As suggested, we have moved the RBC and platelet data from young mice to the main figure to facilitate direct comparison and improve clarity (Fig. 1h,i).

Figure 1f-1i

Fig 1: Could you also add comparisons between OAL and OCRRF? (OAL vs. OCR and OCR vs. OCRRF are shown, presumably OAL vs. OCRRF are similar, but please show for clarity)

Response: We thank the reviewer for this suggestion. Comparisons between OAL and OCRRF have been added as requested, and all relevant figures have been updated in the revised manuscript to reflect these changes.

Figure 1

Figure 2:

Please provide details in text for HSC markers used in RNAseq so readers do not need to search through methods.

Response: We thank the reviewer for this helpful suggestion. The HSC markers (LSK CD34⁻ Flk2⁻ CD150⁺) have been added to the text to provide clarity, allowing readers to understand the populations used for RNA-seq without referring to the Methods section.

Fig 2d: It may be helpful to perform and GSEA comparing OCR (on left) versus OAL (on right) which would yield positive (+) NES (normalized enrichment scores) for cell cycling pathways, and better match the text and Fig. 1c stating that OCR activates cell cycle progression genes. The data is convincing; it would be easier to follow if the enrichment scores match the text and conclusions.

Response: We thank the reviewer for this helpful suggestion. We have regenerated the GSEA comparing OCR (left) versus OAL (right) to ensure that the enrichment scores are consistent with the text (Fig. 2d). In addition, the corresponding GSEA plots for young mice have also been updated in Extended Data Fig. 4d.

Fig. 2d

Extended Data Fig. 4d

Fig. 2g: OCR seems to expand preB –is this significant? If yes, could this be discussed and investigated further? Is there a block in differentiation since B cells are down-regulated?

Response: We thank the reviewer for this insightful comment. The CellRadar tool (<https://karlssong.github.io/cellradar/>) provides a qualitative assessment of relative gene expression enrichment, but it does not allow formal statistical testing to determine significance. We do observe enrichment of pre-B-related genes in OCR HSCs (Fig. 2g), although this enrichment is not apparent in young CR HSCs (Extended Data Fig. 4i). Since young CR mice also show decreased lymphoid differentiation, it is unclear whether the pre-B gene enrichment in OCR HSCs reflects a block in differentiation toward mature B cells. However, we can confirm that CR causes a block in lymphoid differentiation overall: lymphoid cell numbers and frequency decrease after CR, whereas refeeding leads to a rebound in lymphoid production, sometimes exceeding levels seen in ad libitum-fed mice (Fig. 1d, 1l, 1n).

Fig. 2g

Extended Data Fig. 4i

Fig. 1d

Fig. 1l

Fig. 1n

Extended Figure 4:

The trend noted with more down-regulated genes in YCR versus YAL seems significant from the heat maps (Extended Fig 4) – are there other statistical approaches to compare the proportion of up versus down-regulated genes?

Response: We thank the reviewer for this suggestion. We formally tested whether the observed imbalance between up- and down-regulated genes in YCR versus YAL was statistically significant. Among 136 DEGs, 31 were up-regulated and 105 were down-regulated. A two-sided exact binomial test confirmed a significant deviation from a 50:50

distribution ($p = 1.33 \times 10^{-10}$), and a chi-square goodness-of-fit test yielded a very similar result ($p = 2.22 \times 10^{-10}$). These analyses support the observation of a strong bias toward down-regulation in YCR HSCs.

We have added this comparison and conclusion in the revised manuscript:

“We observed a significant bias toward down-regulation of transcripts in YCR versus YAL (31 up vs 105 down of 136 DEGs; two-sided exact binomial test, $p = 1.33 \times 10^{-10}$).”

Figure 3:

Fig. 3a: AS dysregulated genes are shown in a bar graph, but not the specific genes. A table/excel file of these genes and their differential expression and P values would be of interest.

Response: Thank you- In response, we have added Extended Data Table 2, which lists the AS genes shown in Fig. 3a, along with their log2 fold-change (log2FC) and false discovery rate (FDR).

Please discuss the genes *Clu*, *Osmr*, *Ect2*, *Mcm5* further since they were discussed in the text and examined in Fig. 3e, although their full gene names and function may not be well-known to many readers.

Response: As requested, we have added a brief introduction to these four genes in the revised manuscript:

“Examining the AS genes that were differentially expressed in opposite directions in OCR compared to OAL HSCs showed that nearly half of the AS genes exhibited a significant mitigation in expression changes under CR conditions (Fig. 3c). This included *Clu* and *Osmr*, which are upregulated HSC aging signature genes (Fig. 3d). *Clu* (clusterin) encodes a secreted glycoprotein that drives myeloid bias in aged HSCs by regulating mitochondrial function (Sun et al., Nature Aging, 2025, <https://doi.org/10.1038/s43587-025-00908-z>), while *Osmr* (oncostatin M receptor) encodes a cytokine receptor involved in the maintenance of erythroid and megakaryocyte progenitors (Tanaka et al., Blood, 2003, <https://doi.org/10.1182/blood-2003-02-0367>). In contrast, *Ect2* and *Mcm5* are downregulated HSC aging signature genes (Fig. 3e). *Ect2* (epithelial cell transforming 2) encodes a guanine nucleotide exchange factor required for cytokinesis and cell cycle progression, whereas *Mcm5* (*minichromosome maintenance complex component 5*) encodes a subunit of the MCM helicase complex essential for DNA replication initiation and serves as a marker of proliferative potential.”

Figure 4:

The text mentions that sirtuin genes were not altered with CR, although genes regulating histone methylation were reported to be differentially regulated. Please elaborate further and show the genes and data supporting this.

Response: We thank the reviewer for this helpful suggestion. In response, we have added Extended Data Table 3, which lists the expression levels of all sirtuin genes as well as histone methylation regulators. As shown in the table, sirtuin genes did not display significant changes under CR, whereas two H3K27me3 regulators, Ezh1 and Ezh2, were differentially expressed.

Gene	log2FoldChange	padj
Sirt1	-0.037549716	0.919639983
Sirt2	-0.073766395	0.737278393
Sirt3	-0.228080983	0.105453887
Sirt4	-0.08467299	0.803382919
Sirt5	-0.048333345	0.868897668
Sirt6	-0.086887848	0.778467115
Sirt7	0.01697516	0.957540427
Kmt2a	0.035214198	0.913022708
Kmt2b	0.106769233	0.692049507
Kmt2c	0.045983322	0.863611208
Kmt2d	-0.003789393	0.990367801
Setd1a	-0.005018577	0.989375308
Setd1b	-0.186822566	0.387418565
Ezh1	-0.469815351	0.000403736
Ezh2	0.59945768	5.45E-06

Please also provide rationale for focusing on H3K4me3 and H3K427me3. Did the gene expression results suggest these marks would be affected?

Fig. 4c-i: Gene silencing experiments suggest that KDR is a key regulator of the CR response and effects on aging signature.

Response: We focused on H3K4me3 and H3K27me3 because these histone marks are tightly linked to transcriptional regulation: H3K4me3 is enriched at active promoters, whereas H3K27me3 mediates transcriptional repression. Gene expression analysis of enzymes associated with H3K4me3 and H3K27me3 suggested potential dysregulation of these modifications in HSCs under CR conditions. Specifically, in OCR HSCs we observed an unexpected shift in the expression of H3K27me3 methyltransferases - Ezh1 was decreased and Ezh2 increased - which contrasts with the pattern seen during physiological aging, where Ezh1 increases and Ezh2 decreases. However, changes in enzyme expression

alone cannot reveal which genes are directly affected or the nature of those changes. Therefore, we performed H3K4me3 and H3K27me3 ChIP-seq to define locus-specific alterations, enabling us to pinpoint the genomic targets of these histone modifications and to identify potential regulators of the CR response.

Text Line 200 – Please clarify: Alternatively, we could have examined the 7 genes with increased H3K4me3 that mitigated the decreased expression associated with aging in HSC. I would either examine these genes or not mention that you could examine them.

Response: We apologize for this poor phrasing. We agree that the original sentence was potentially confusing. Accordingly, we have removed this sentence in the revised manuscript.

Figure 5:

Based on ATACseq, PU.1 is implicated as a primary regulator of the CR response which was validated by ChIPseq. What is the link between KDR signaling and PU.1? Could experiments or bioinformatic analyses be performed to identify potential pathways that link these 2 factors?

Response: We thank the reviewer for this insightful question regarding the potential connection between KDR signaling and PU.1. To explore this link, we analyzed whether genes differentially expressed after Kdr knockdown were regulated by PU.1. Approximately 50% of the DEGs showed PU.1 binding within 50 kb of the transcription start site (TSS) (Extended Data Fig. 8b). Notably, chromatin profiling revealed that PU.1 binding increased after CR (Extended Data Fig. 8c), suggesting that PU.1 participates in the transcriptional regulation of KDR-responsive genes under CR conditions.

Further analysis indicated that both CR and Kdr knockdown led to a modest increase in Spi1 expression (Extended Data Fig. 8d), and ChIP-seq data showed PU.1 occupancy near the Kdr promoter (Extended Data Fig. 8e), supporting a possible reciprocal regulatory interaction between KDR signaling and PU.1 activity. However, GSEA did not detect significant enrichment of canonical KDR-related pathways after CR (Table R1), implying that the transcriptional impact of KDR signaling perturbation may be limited, as KDR primarily signals through phosphorylation cascades.

Extended Data Fig. 8

Table R1

NAME	NES	FDR q-val
BIOCARTA_AKT_PATHWAY	1.0391848	0.8805517
BIOCARTA_MAPK_PATHWAY	1.0423836	0.8828368
BIOCARTA_P38MAPK_PATHWAY	-0.57363373	1
BIOCARTA_STAT3_PATHWAY	-0.5297276	1

Some additional issues to address:

The findings that CR increases HSC cycling with a myeloid bias while mitigating AS genes seems contradictory. How does the LT-HSC in OCR compare to Young – is CR restraining HSC expansion that occurs with aging?

How can we reconcile increased HSC cycling and myeloid bias (2 aging phenotypes) with dampening of the AS? What are the specific AS genes?

Are the quiescent LT-HSC being preserved with CR? Examination of quiescent (CD41 low/Procr high and/or other markers of quiescence) versus cycling LT-HSC (CD41 high/Procr low and/or other markers of cycling) could provide further insight into how CR represses AS while promoting cycling and myeloid bias.

Response: Thank you for this insightful question. To clarify this seemingly contradictory observation, we analyzed CD150-high (myeloid-biased) and CD150-low (balanced) HSC subpopulations, as the frequency of cycling HSCs (CD41 high/Procr low) is higher in CD150-high HSCs compared to CD150-low HSCs (Fig. R1a). After CR, the frequency of CD150-high HSCs decreased, while CD150-low HSCs were maintained (Extended Data Fig. 3h), suggesting that CR primarily affects myeloid-biased HSCs, which may cycle more and differentiate under CR stress. Following refeeding, CD150-high HSCs returned to OAL levels, whereas CD150-low HSCs decreased (Extended Data Fig. 3h), which may reflect a burst of lymphoid differentiation. As CD150-low HSCs retain a younger transcriptome (Wang et al. 2025, Cell Research, <https://doi.org/10.1038/s41422-024-01057-5>), this shift likely contributes to the improved aging transcriptome observed in OCR HSCs.

Overall, the transcriptome of OCR HSCs trends toward young HSCs but is not fully restored (Fig. R1b). CR partially restrains age-associated HSC expansion, though the population still increases, albeit to a lesser extent than in ad lib fed aged HSCs (Fig. R1c).

We have also added Extended Data Table 2, which lists all the AS genes, together with their \log_2 fold-change (\log_2FC) and false discovery rate (FDR).

Fig. R1b**Fig. R1c**
Along these lines, are the expanded HSC acquiring mutations as occurs with human HSC aging? Although this is beyond the scope of this paper, could the authors speculate on whether CR may modulate acquisition of mutations that allow HSC expansion?

Response: DNA replication is a major source of mutations, and CR increases HSC cycling, which could potentially influence mutation acquisition. We examined the RNA-seq data for mutations in the reported clonal hematopoiesis (CH) genes which acquire mutations associated with human aging, and do not see an increase in mutations in the CR HSCs. Our data also indicate that the cycling HSCs under CR mainly differentiate to maintain the myeloid lineage rather than self-renew. Therefore, we speculate that while CR may modulate mutation acquisition in HSCs, we don't see an increase in mutation rate of the CH genes and posit that it is unlikely that acquisition of these mutations would result in substantial changes in murine HSC expansion in this experimental paradigm.

While beyond the scope of this paper, transcriptomic analyses of spleens and spleen/bone marrow single cell analyses would provide further insight into mechanisms underlying CR with aging.

Response: We thank the reviewer for this thoughtful suggestion. We agree that transcriptomic analyses of spleen, as well as single-cell analyses of spleen and bone marrow, could provide additional insight into the mechanisms underlying CR with aging, and we will consider these experiments in future studies.

Reviewer #2 (Remarks to the Author):

Summary

This manuscript provides a mechanistically detailed and well-executed investigation into how caloric restriction (CR) remodels the aging hematopoietic stem cell (HSC) compartment. The authors report that CR promotes a myeloid-biased transcriptional program at the expense of lymphoid differentiation, driven in part by epigenetic silencing of *Kdr* and *Bmpr1a* and increased PU.1 chromatin occupancy. The findings are timely and impactful; however, several areas require further clarification or additional data to strengthen the manuscript's claims and contextualization.

Response: We thank the reviewer for the thoughtful and positive evaluation of our work and for recognizing the impact of our study. We appreciate the constructive feedback highlighting areas for clarification and additional context. In the revised manuscript, we have carefully addressed all comments point by point. We have provided additional analyses, expanded data presentation, and refined the discussion to better contextualize our findings and strengthen the overall conclusions.

Specific suggestions:

Definition of "Youthfulness"

While the manuscript asserts that CR preserves a youthful HSC transcriptome, the observed phenotype is also characterized by reduced lymphoid output—particularly naive T cells—which are hallmarks of youth. The current phenotype may instead reflect a regulated or functionally adaptive response to energetic stress. I recommend refining the language throughout to differentiate between transcriptional rejuvenation and functional youthfulness. A deeper analysis of which aged HSC genes are reversed by CR and which persist would clarify the extent of this rejuvenation.

Response: We thank the reviewer for this important comment. We agree that our data support transcriptional rejuvenation rather than functional youthfulness. Accordingly, we have revised the manuscript to use “transcriptional rejuvenation” throughout.

To further clarify the extent of this rejuvenation, we added Extended Data Table 2, which lists all HSC aging signature (AS) genes together with their \log_2 fold-change ($\log_2\text{FC}$) and false discovery rate (FDR). Among these, 111 genes were reversed by CR (fold-change > 1.2, FDR < 0.05), whereas 98 genes persisted (fold-change < 1.2 or FDR > 0.05). These data delineate the subset of aging-associated transcriptional changes that are responsive to CR versus those that remain unaffected.

Direct comparison across age and CR groups

Although the authors refer to a youthful signature, they do not show a PCA or clustering analysis that includes both young ad libitum (YAL) and young CR (YCR) HSCs alongside old CR (OCR) HSCs. Including these groups in PCA or trajectory analyses would allow the readers to evaluate whether OCR HSCs genuinely converge toward a youthful transcriptional profile or instead form a unique CR-adapted state.

Response: We thank the reviewer for this valuable suggestion. As recommended, we performed PCA and unsupervised clustering analyses including YAL, YCR, OAL, and OCR HSCs. The PCA showed that lifelong CR (OCR) shifted the transcriptome toward the young HSC (YAL) state along PC1 (Extended Data Fig. 5a). In the unsupervised clustering analysis, YAL and YCR grouped together, while OAL and OCR formed a separate cluster (Extended Data Fig. 5b). Nonetheless, the transcriptomic profile of OCR HSCs clearly shifted toward, but did not fully converge with, the YAL state, indicating a partial transcriptional reversion under CR.

Extended Data Fig. 5

Clarification of CR protocol

More detail is needed on the CR paradigm, particularly concerning the weekend feeding regimen. Were mice subjected to continuous daily restriction, or did they receive multi-day rations? Such patterns could lead to unintentional refeeding-fasting cycles, influencing immune and HSC outcomes. Clarifying whether CR was continuous or intermittent is important for interpreting discrepancies with previous reports and for translational relevance.

Response: We appreciate the reviewer’s critical comment regarding the CR paradigm. In our study, CR was implemented as a continuous regimen, with mice receiving the same restricted feeding both on weekdays and weekends. Thus, the animals were not exposed to intermittent refeeding-fasting cycles. We have revised the Methods section to clearly describe this protocol.

T cell output

The authors quantify CD3+ cells in their gating strategy, yet T cell data are not reported. This omission is notable, as it precludes comparison with studies (e.g., Di Francesco et al. Nature 2024) that observed CR-mediated preservation of T cells. Reporting absolute and relative T cell numbers would help resolve whether CR compromises adaptive immunity broadly or selectively suppresses B cells.

Response: We thank the reviewer for this valuable comment. We have now included both the absolute number and relative frequency of CD3+ T cells in the revised data. Our analysis shows that both short-term and lifelong CR significantly reduce the absolute number of CD3+ T cells (Extended Data Fig. 2i), while the relative frequency of T cells remains largely unchanged (Extended Data Fig. 2h). In contrast, CR decreases both the absolute number and frequency of B cells (Fig. 1l and Extended Data Fig. 2g, 2j), indicating that B cells are more selectively suppressed than T cells under CR conditions.

Extended Data Fig. 2i

Extended Data Fig. 2h

Extended Data Fig. 2j

Extended Data Fig. 2g

Fig. 1l

Functional evidence for immune suppression

The claim that CR may compromise immune function would be strengthened by functional validation, such as infection or tumor challenge models. In the absence of these experiments, the authors should clearly acknowledge this limitation and revise language in the abstract and discussion to avoid overinterpretation.

Response: We thank the reviewer for this important comment. We fully agree that functional validation, such as infection or tumor challenge models, would provide stronger evidence for the impact of CR on immune competence. We acknowledge that such experiments were not performed in the current study and have now explicitly stated this limitation in the revised Discussion.

To provide context, previous studies have indeed demonstrated impaired immune responses under CR conditions. For example, Tang et al. reported that CR led to reduced bacterial clearance following infection (Tang et al., *J. Exp. Med.*, 2016, <https://doi.org/10.1084/jem.20151100>). Similarly, Kristan et al. found that CR increased susceptibility of laboratory mice to intestinal parasite infection (Kristan, *Aging Cell*, 2007, <https://doi.org/10.1111/j.1474-9726.2007.00345.x>), and Gardner et al. showed that CR decreased survival, increased viral titers, and reduced natural killer cell activity following influenza infection in aged mice (Gardner, *J. Gerontol. A*, 2005, <https://doi.org/10.1093/gerona/60.6.688>).

In line with the reviewer's suggestion, we have revised the Abstract and Discussion to temper our interpretation. We now describe that "CR has been reported to be associated with reduced immune function in certain contexts" rather than stating directly that "CR compromises immune function." We have also added a sentence in the Discussion explicitly acknowledging the absence of direct functional validation of immune suppression in our study.

Biological replicates in transcriptomic analyses

In Figure 2a, only n=2 samples are included for the OCRRF group, while other groups have n=4. The authors should explain why OCRRF replicates were reduced and clarify whether conclusions drawn from this group are robust. If possible, additional replicates should be included.

Response: We thank the reviewer for this valuable comment. We initially included four mice per group. However, upon dissection, we found that two mice in the OCRRF group presented with health abnormalities - one had a large liver tumor and the other had an abnormally small spleen. Because these conditions could introduce confounding effects unrelated to our

experimental variables, we did not proceed with RNA-seq for these two animals, resulting in n=2 for the OCRRF group.

To assess the robustness of our conclusions, we later performed RNA-seq on additional OCRRF mice. The transcriptomic profiles of these new samples were consistent with those of the original OCRRF replicates (Fig. 2a,b and Fig. R2), supporting the reliability of our findings. These additional data are provided to the reviewers for reference.

Because RNA-seq data generated at different times can be influenced by batch effects, we present only samples processed and sequenced within the same experimental batch to ensure internal consistency across all groups. Importantly, the inclusion of additional OCRRF replicates did not alter the key findings, supporting the robustness of the conclusions despite the reduced sample size in the original dataset.

Fig. 2a

Fig. 2b

Fig. R2 (with 2 additional OCRRF)

Bone marrow cellularity and absolute counts

The manuscript reports frequencies of HSCs and progenitor populations but omits data on total BM cellularity. Without absolute counts, it is difficult to distinguish between genuine expansion/contraction of cell populations and shifts due to altered gating proportions. Including BM cellularity and absolute numbers of cell populations would substantially strengthen the phenotypic conclusions.

Response: Thank you for this comment. We agree that bone marrow (BM) cellularity is important for distinguishing genuine expansion/contraction of cell populations and shifts due to altered gating proportions.

To address this concern, we quantified total BM cellularity and found that overall BM cellularity remained largely unchanged after CR, with no statistically significant differences between groups (Extended Data Fig. 3i). This contrasts with peripheral blood, where total white blood cell counts were significantly reduced under CR conditions (Extended Data Fig. 2a).

These findings suggest that while CR alters the composition of hematopoietic populations - reducing B cell frequencies and increasing myeloid frequencies in both BM and blood - the total number of BM cells is maintained. Thus, the observed shifts in BM lineage frequencies primarily reflect changes in lineage balance rather than overall cell loss. This distinction highlights a key difference between the bone marrow and peripheral blood in their responses to CR.

Extended Data Fig. 3i

Extended Data Fig. 2a

Kdr ligand expression

The authors should evaluate whether expression of Kdr ligands (e.g., VEGFs) is altered with aging or CR, either in their dataset or via comparison with public datasets. This would strengthen the case for Kdr as a physiologically relevant modulator of HSC aging.

Response: We thank the reviewer for this valuable suggestion. In mice, three known ligands bind to Kdr: *Vegfa*, *Vegfc*, and *Vegfd*. In our dataset, *Vegfa* and *Vegfc* were readily detected in HSCs, whereas *Vegfd* expression was very low (Fig. R3).

With aging, Kdr expression increased, accompanied by a decrease in *Vegfa* and no substantial change in *Vegfc* (Fig. R3). Under lifelong CR, Kdr expression was reduced compared to aged controls, while *Vegfa* showed a modest trend toward increased expression and *Vegfc* showed a trend toward decreased expression (Fig. R3). These patterns suggest that CR may mitigate age-associated upregulation of Kdr and subtly modulate its ligand balance. Nonetheless, Kdr signaling in HSCs may also be influenced by ligands derived from niche or systemic sources.

Fig. R3

PU.1 and Kdr Pathway Integration

The manuscript presents parallel evidence for PU.1 and Kdr as CR response regulators but does not clarify whether these pathways intersect mechanistically. Is there evidence that PU.1 binding is altered at loci with CR-induced histone changes? Are any PU.1-bound genes among those regulated by Kdr? Exploring overlap in differential chromatin accessibility and histone methylation could uncover a more unified mechanism.

Response: We thank the reviewer for this insightful question regarding the potential mechanistic integration between PU.1 and Kdr pathways. We examined the genomic distribution of PU.1 binding sites, regions of altered chromatin accessibility (DARs), and loci with CR-associated histone modification changes. PU.1 binding and DARs were

predominantly located at distal regulatory elements rather than promoters, whereas changes in histone modifications (H3K4me3, H3K27me3) were mainly confined to promoter regions. Consequently, there was minimal direct overlap between PU.1 binding/DARs and promoter-associated histone changes. However, such non-overlap does not exclude functional interaction, as PU.1-bound enhancers can still regulate genes whose promoters show histone modification changes through enhancer–promoter interactions.

To further explore the potential link between these pathways, as the reviewer suggested, we analyzed whether genes differentially expressed following Kdr knockdown were regulated by PU.1. Approximately 50% of these DEGs had PU.1 binding within 50 kb of the transcription start site (TSS) (Extended Data Fig. 8b), indicating substantial potential regulatory overlap. Moreover, chromatin profiling revealed increased PU.1 binding after CR (Extended Data Fig. 8c), suggesting that PU.1 contributes to the transcriptional regulation of KDR-responsive genes under CR conditions.

We also found that both CR and Kdr knockdown led to a modest increase in Spi1 expression (Extended Data Fig. 8d), and ChIP-seq data showed PU.1 occupancy near the Kdr promoter (Extended Data Fig. 8e), supporting a possible reciprocal regulatory interaction between KDR signaling and PU.1 activity.

Extended Data Fig. 8

b

c

PU.1 Binding Partners

For genes with differential PU.1 occupancy, are different adjacent transcription factor motifs enriched among upregulated vs downregulated genes? This could suggest context-specific co-factors that shape PU.1's dual roles as an activator and repressor. This analysis would provide important mechanistic depth.

Response: We thank the reviewer for this thoughtful suggestion. To address this, we classified PU.1 peaks showing altered occupancy into two groups according to whether they were associated with genes upregulated or downregulated after CR. We then performed transcription factor binding motif enrichment analysis using HOMER, analyzing sequences within ± 200 bp of the PU.1 peak centers. Both *de novo* and known motif analyses were conducted.

As expected, the PU.1 motif was the top enriched motif in both groups, confirming the specificity and robustness of the dataset (Extended Data Fig. 8a). Among known motifs, we observed similar enrichment of ETS family transcription factors - including ELF4, ETS1, FLI1, GABPA, and ERG - across both up- and downregulated gene-associated peaks (Extended Data Fig. 8a, left). These factors share the ETS DNA-binding domain and are known to act as either activators or repressors in a context-dependent manner, consistent with PU.1's dual regulatory roles.

Interestingly, *de novo* motif analysis revealed modest enrichment of distinct motifs between the two gene sets: NFIA and HMGA1 motifs were more enriched in PU.1 peaks linked to upregulated genes, whereas ZFP335 and MYBL2 motifs were more enriched in those associated with downregulated genes, although their enrichment p-values were substantially higher than that of PU.1 (Extended Data Fig. 8a, right). These results suggest that PU.1 may engage different co-regulatory partners in distinct transcriptional contexts. Alternatively, PU.1's dual activity may involve sequence-independent mechanisms, such as recruitment of different chromatin modifiers to regulate transcription.

Extended Data Fig. 8a

Minor points

Currently the text mentions B220+ cells (line 93). Is B220 used as a marker for B cells? If so, the text should state that there is decrease in B cells with both short-term and long term CR for clarification.

Response: We thank the reviewer for this helpful comment. Yes, B220 is used as a marker for B cells. As suggested, we have revised the text to replace “B220+ cells” with “B cells” to clarify that B cell frequency decreases under both short-term and lifelong CR conditions.

OWT is used in the text starting on line 258. This most likely refers to the Old Ad Lib (OAL) mice and should be corrected for consistency.

Response: We thank the reviewer for catching this inconsistency. We have corrected “OWT” to “OAL” throughout the revised manuscript, including both the text and figure labels, to ensure consistency.

Reviewer #3 (Remarks to the Author):

In this manuscript, Zong et al. investigate the mechanisms underlying the effects of caloric restriction (CR) on the immune function of aged mice. They first confirm that CR leads to reduced white blood cell counts and myeloid bias. They then examined the effects of short term and long term CR as well as refeeding on transcription aging signatures in hematopoietic stem cells (HSCs) and demonstrate that CR preserves the 'youthful' signature. They examined epigenetic changes in specific histone methylation marks which identified Kdr and Bmpr1a as candidate regulators of the CR effects. Subsequent knockdown experiments confirmed that Kdr is able to recapitulate the CR effect. Examination of differential open chromatin regions further implicated Pu.1 as a candidate inhibitor of HSC renewal.

This work substantially advances our understanding the molecular mechanisms underlying immune response to CR.

The experiments are clearly described and follow a logical progression that supports the conclusions.

My only major concern is that the study is limited to male C57BL/6J mice. Including female mice should be considered essential, despite previous studies reporting absence of CR effects. Further the use of a single genetic background limits the scope of the conclusions as it is unclear if these responses are universal or limited to (male) C57 mice. Nonetheless the study is valuable and advances new hypotheses that can be explored in other contexts.

Response: We thank the reviewer for the positive evaluation of our work and for highlighting this important point. We fully agree that including female mice and additional genetic backgrounds would strengthen the generalizability of our conclusions. We recognize that sex differences play a significant role in the aging of the hematopoietic system. In fact, we are currently preparing a separate study focused on sex-dependent changes in hematopoiesis during aging. We will also examine the effects of CR in female mice in future investigations to determine whether the mechanisms identified here are conserved across sexes.

Reviewer #4 (Remarks to the Author):
